# Ubiquitin-Specific Protease 6 n-Terminal-like Protein (USP6NL) and the Epidermal Growth Factor Receptor (EGFR) Signaling Axis Regulates Ubiquitin-Mediated DNA Repair and Temozolomide-Resistance in Glioblastoma

**DOI:** 10.3390/biomedicines10071531

**Published:** 2022-06-28

**Authors:** I-Chang Su, Yu-Kai Su, Hao-Yu Chuang, Vijesh Kumar Yadav, Syahru Agung Setiawan, Iat-Hang Fong, Chi-Tai Yeh, Hui-Chuan Huang, Chien-Min Lin

**Affiliations:** 1Department of Neurology, School of Medicine, College of Medicine, Taipei Medical University, Taipei City 11031, Taiwan; ichangsu@gmail.com (I.-C.S.); yukai.su@gmail.com (Y.-K.S.); vijeshp2@gmail.com (V.K.Y.); impossiblewasnothing@hotmail.com (I.-H.F.); 2Division of Neurosurgery, Department of Surgery, Taipei Medical University-Shuang Ho Hospital, New Taipei City 23561, Taiwan; 3Taipei Neuroscience Institute, Taipei Medical University, Taipei 11031, Taiwan; 4Graduate Institute of Clinical Medicine, College of Medicine, Taipei Medical University, Taipei City 11031, Taiwan; 5Cell Therapy Center, Tainan Municipal An-Nan Hospital, China Medical University, Tainan 709, Taiwan; greeberg1975@gmail.com; 6Division of Neurosurgery, Tainan Municipal An-Nan Hospital, China Medical University, Tainan 709, Taiwan; 7Division of Neurosurgery, China Medical University Beigang Hospital, Yunlin 65152, Taiwan; 8International Ph.D. Program in Medicine, College of Medicine, Taipei Medical University, Taipei City 11031, Taiwan; setiawan.syahru@gmail.com; 9Department of Medical Research & Education, Shuang Ho Hospital, Taipei Medical University, New Taipei City 23561, Taiwan; 10College of Nursing, Taipei Medical University, Taipei 11031, Taiwan; huichuan@tmu.edu.tw

**Keywords:** glioblastoma, ubiquitin-proteasome system, DNA repair, drug resistance, USP6NL

## Abstract

Glioblastoma multiforme (GBM) is the most malignant glioma, with a 30–60% epidermal growth factor receptor (EGFR) mutation. This mutation is associated with unrestricted cell growth and increases the possibility of cancer invasion. Patients with EGFR-mutated GBM often develop resistance to the available treatment modalities and higher recurrence rates. The drug resistance observed is associated with multiple genetic or epigenetic factors. The ubiquitin-specific protease 6 N-terminal-like protein (USP6NL) is a GTPase-activating protein that functions as a deubiquitinating enzyme and regulates endocytosis and signal transduction. It is highly expressed in many cancer types and may promote the growth and proliferation of cancer cells. We hypothesized that USP6NL affects GBM chemoresistance and tumorigenesis, and that its inhibition may be a novel therapeutic strategy for GBM treatment. The USP6NL level, together with EGFR expression in human GBM tissue samples and cell lines associated with therapy resistance, tumor growth, and cancer invasion, were investigated. Its pivotal roles and potential mechanism in modulating tumor growth, and the key mechanism associated with therapy resistance of GBM cells, were studied, both in vitro and in vivo. Herein, we found that deubiquitinase USP6NL and growth factor receptor EGFR were strongly associated with the oncogenicity and resistance of GBM, both in vitro and in vivo, toward temozolomide, as evidenced by enhanced migration, invasion, and acquisition of a highly invasive and drug-resistant phenotype by the GBM cells. Furthermore, abrogation of USP6NL reversed the properties of GBM cells and resensitized them toward temozolomide by enhancing autophagy and reducing the DNA damage repair response. Our results provide novel insights into the probable mechanism through which USP6NL/EGFR signaling might suppress the anticancer therapeutic response, induce cancer invasiveness, and facilitate reduced sensitivity to temozolomide treatment in GBM in an autolysosome-dependent manner. Therefore, controlling the USP6NL may offer an alternative, but efficient, therapeutic strategy for targeting and eradicating otherwise resistant and recurrent phenotypes of aggressive GBM cells.

## 1. Introduction

Glioblastoma (GBM), a group of poorly differentiated aggressive primary astrocytic or oligodendroglial tumors, is one of the most frequently diagnosed central nervous system (CNS) malignancies worldwide, with a high mortality rate [1]. Despite advances in diagnostics and therapeutics, most patients with GBM are not sensitive to the current treatment modalities, or show relapse/progress after initial response to therapy, thus rendering GBM incurable; the median survival time is only 12–15 months [1,2,3]. GBM is characterized by different genetic alterations, such as mutations in the isocitrate dehydrogenase-1 or -2 genes, methylation of the O6-methylgunine-DNA methyltransferase (MGMT) promoter, and overexpression of the gene encoding the epidermal growth factor receptor (EGFR) [4,5,6,7]. Isocitrate dehydrogenase and MGMT promoter status have better prognostic significance in predicting GBM patient survival after therapy; however, the role of EGFR in the survival of patients with GBM remains controversial. The high mortality rate, treatment resistance, and tendency to relapse necessitate the identification of new druggable molecular targets or the development of novel effective precise therapeutics for patients with GBM.

Besides EGFR, cell cycle regulation plays a pivotal role in the determination of cell fate, especially cancerous cells [8]. A dysregulated cell cycle and checkpoint disruption are crucial for the survival, development, progression, and chemoresistance of malignant cells; in fact, impaired cell cycle regulation affects cell growth, differentiation, and response to stress or death signals, all of which are hallmarks of cancer [9,10]. Thus, an understanding of the mechanisms underlying cell cycle regulation is critical for the development of novel and efficacious therapeutic targets.

Ubiquitination is a posttranslational modification process that regulates both physiological and pathological biocellular activities [11]. Through the addition of the 76 amino acid long ubiquitin or ubiquitin-like proteins to target proteins, they are flagged for proteasomal degradation [12]. The ubiquitin-proteasome system (UPS), cellular machinery for protein tagging and proteolytic degradation, comprises ubiquitin; 26S proteasome; and the ubiquitin-activating (E1), ubiquitin-conjugating (E2), and ubiquitin-ligase (E3) enzyme complex, and it is actively involved in various structural and functional intracellular activities [12]. The UPS presents a promising field for novel anti-GBM therapeutics [13]. Every enzymatic component of the UPS and deubiquitinases may be targetable [13,14]. This represents a vast pool of candidate therapeutic targets; additionally, exploiting protein turnover may represent a novel treatment strategy for targeting the hitherto “undruggable proteome” [13,14,15]. Drug resistance is induced by many internal factors, such as the induction of drug efflux, decreased drug uptake, alteration in cell cycle checkpoints, induced expression of emergency response genes, inhibition of apoptosis, autophagy, alteration of the drug target, and drug compartmentalization [16].

In malignant cells, many cellular proteins, regardless of their intracellular roles, including cell cycle control proteins, are directly targeted by impaired degradation machinery in the context of dysregulated UPS-regulated signaling pathways [12,17], making the UPS a critical factor in cell cycle regulation. Besides surgical resection, ionizing radiation, which damages the DNA of cancerous cells, and temozolomide (TMZ), a DNA alkylating agent that elicits base pair mismatch and futile DNA repair, remain the mainstays of managing patients with GBM [17]. Loss of the ability to repair DNA damage will lead to mutations, which induces therapy resistance in cancer cells [18]. Consequently, biologically vital feedback control exists between cell cycle control and DNA repair disrupts [19]. The maintenance of genomic stability depends on the precise coherence or balance between DNA replication, chromosome segregation, and DNA repair, integrated with cell cycle progression and several other processes, including ubiquitination [12,17,18,19,20]. This preserved genomic stability is critical for cellular homeostasis and the suppression of tumors. In cancer, especially non-small cell lung cancer (NSCLC), the higher burden in genes encoding proteins such as EGFR, KRAS, TP53, MYC, and ALK contributes to genomic instability [21,22]. Therefore, gaining an understanding of mechanistic insight and identifying potential pharmacological targets in the improvement of current treatment is vital.

Ubiquitin-specific peptidase 6 N-terminal like (USP6NL, also called RN-tre or Tre2) has been implicated in some human malignancies, namely gastric cancer, colorectal carcinoma, and breast cancer [23,24,25]. USP6NL acts as a GTPase-activating protein for Ras-related protein, and is involved in receptor trafficking. It inhibits EGFR internalization in a complex with EPS8 and decreases Rab5 activity [25]. A substantial amount of evidence supports the finding that USP6NL inhibits EGFR endocytosis. USP6NL increases the constitutive internalization of EGFR and α5β1 integrins, which suggests a role for USP6NL in the EPS8-constrained endocytosis of EGFR, as well as the unstable organization of adhesion complexes and the dissociation of the EGF-dependent adhesion complex [25,26]. Breast cancer cells with high levels of USP6NL experienced delayed endocytosis and degradation of EGFR, activated AKT signaling. Deficiency of USP6NL caused downregulation of EGFR, resulting in the suppression of AKT and GLUT1 degradation and the impairment of cellular proliferation [27]. Thus, we hypothesized that USP6NL regulates not only EGFR trafficking, but also the cellular survival capacity or bio-availability of adhesion and growth factor receptors, including EGFR. Consistent with the suggestion that targeting USP6NL enhanced β-catenin ubiquitination suppressed cancer cell proliferation and induced cell cycle arrest in colorectal cell lines [23], as well as considering that EGFR is the most commonly mutated driver of oncogenicity in patients with GBM, there was hope for targeting EGFR with inhibitors; unfortunately, this showed no or very little response toward treatment [28].

Therefore, we explored the interplay between USP6NL, together with known EGFR status effectors of TMZ resistance, DNA damage repair, autophagy, and many mediators of GBM tumorigenesis, which may offer an alternative but efficient therapeutic strategy for targeting and eradicating otherwise resistant and recurrent phenotypes of aggressive GBM cells.

## 2. Materials and Methods

### 2.1. GBM Cell Culture

Hs683, T98G, DBTRG05MG, and U87MG GBM cell lines were obtained from the American Form Culture Collection (ATCC; Manassas, VA, USA) and maintained in an incubator with 5% CO_2_ in humidified air. The cells were cultured in Dulbecco’s modified Eagle’s medium (#12491023; GIBCO, Life Technologies, Carlsbad, CA, USA) supplemented with 10% fetal bovine serum (GIBCO, Life Technologies), penicillin (100 IU/mL), and streptomycin (100 g/mL) (#15140122, GIBCO, Life Technologies). TMZ-resistant U87MGR and U251R cells were generated using a TMZ dose-escalation method up to 150 µM and then maintained at 100 µM TMZ for in vitro and in vivo experiments, according to the methods in earlier references [29,30].

### 2.2. Bioinformatic Analysis of the TCGA-GBM Database

TGCA-GBM mRNA expression data, along with clinical information, were extracted and integrated with corresponding microarray data, including those of GBM and normal CNS tissues (*n* = 370). Extracted clinical profiles include patients’ age, sex, ethnicity, follow-up duration (days), endpoint/event, method of initial confirmed diagnosis, histological type, EGFR mutation status, pathological stage, and grade. The study was approved by the Joint Institutional Review Board (JIRB) of the Taipei Medical University Shuang-Ho Hospital (Approval no. JIRB N202101069). A total of 60 tissue samples from patients with primary and recurrent GBM were obtained from the Taipei Medical University-Shuang-Ho Hospital GBM cohort, in compliance with the recommendations of the Declaration of Helsinki for Biomedical Research. Tissue samples were employed for further immunohistochemistry staining and analysis. Appendix A shows the baseline clinical characteristics and outcomes of GBM patients in SHH-GBM cohort.

### 2.3. shRNA Transfection of GBM Cells

Both parental and resistant clones of U87MG and T98G cells were transfected with shRNA specifically targeting USP6NL, or control/scramble shRNA, purchased from Santa Cruz Biotechnology (Santa Cruz, CA, USA). The U87MG and T98G cells were transfected with shRNA, following the manufacturer’s instructions. shRNA-USP6NL-transfected clones were then expanded for future use.

### 2.4. Cell Viability Assay

The CCK-8 assay was applied for evaluating the cellular viability of adherent and spheroid GBM exposed to the indicated TMZ dose for 24–48 h.

### 2.5. Western Blotting

After all the corresponding treatments, proteins were extracted from the GBM cells and lysed after trypsinization. After the protein lysates were heated, immunoblotting was performed with 5% skimmed milk in Tris-buffered saline with Tween 20 (TBST) for 1 h and then incubated overnight at 4 °C with primary antibodies against the protein of interest (Appendix A). After incubation with the primary antibody, the polyvinylidene difluoride membranes were washed several times with TBST and then incubated for 1 h at room temperature with an HRP-labeled secondary antibody and rewashed with TBST. Subsequently, enhanced chemiluminescence, Western blotting, and a BioSpectrum Imaging System (UVP; Upland, CA, USA) were used to detect the bands.

### 2.6. Immunohistochemistry Staining

Immunohistochemistry staining was performed on formalin-fixed paraffin-embedded sections by using antibodies against USP6NL. Briefly, tissue sections (4 µm) were created and deparaffinized. Following antigen retrieval, blocking solution was applied, then slices were incubated with primary antibodies against USP6NL (Creative Diagnostics Cat# DCABH-17363, RRID: AB_2489260) with a dilution ratio of 1:100 for 2 h at room temperature. After incubation with Horseradish peroxidase (HRP) and Diaminobenzidine (DAB) chromogen, along with substrate, slides were counterstained with hematoxylin, and final mounting solution was applied. As a negative control, a comparable staining method was employed using isotype rabbit IgG for the primary antibody. Cohen’s Kappa values for each pathologist, expressing a quick score of USP6L in glioma tissue, are shown in Appendix A. Two independent pathologists assessed and scored USP6NL expression using the quick-score (Q-score), which is derived according to staining intensity (I) and the proportion of stained cells (P), as previously described [31]. There were four different levels of staining intensity: 0 (no staining), 1+ (weak), 2+ (moderate), and 3+ (strong). Therefore, we finally counted the score as Q = I × P, with a maximum score of 300. Representative staining of USP6L and the control in glioma FFPE tissue sections is shown in Appendix A.

### 2.7. Immunofluorescence Staining

Representative human glioblastoma cell lines harboring either scramble control or the knockdown of USP6L (shUSP6L) were plated in six-well chamber slides for 24 h to perform immunofluorescence analysis. The cells were fixed with 2% paraformaldehyde and probed with primary antibodies. To determine the positive signal, a fluorophore-conjugated secondary antibody was added following examination using a Zeiss Axiophot (Carl Zeiss) fluorescence microscope. The nuclei of viable cells were detected through 4′,6-diamidino-2-phenylindole (DAPI) staining. The primary antibodies used were purchased from several different companies and diluted under specific concentrations, such as E-cadherin (1:100, #3195, rabbit mAb; Cell Signaling, Danvers, MA, USA), N-cadherin (1:100, #13116, rabbit mAb; Cell Signaling), yH2AX (1:100, #7631, rabbit mAb; Cell Signaling), RAD51 (1:100, ab63801, rabbit polyclonal-Ab; Abcam, Cambridge, UK), BRCA2 (1:100, #PA5-96128, rabbit polyclonal-Ab; ThermoFisher, Waltham, MA, USA), ATG5 (1:100, ab228668, rabbit polyclonal-Ab; Abcam), ATG7 (1:100, ab133528, rabbit mAb; Abcam), Beclin-1 (1:100, ab62557, rabbit polyclonal-Ab; Abcam), and LC3A/B (1:100, #4108, rabbit polyclonal-Ab; Cell Signaling). The negative controls were carried out by omitting the primary antibody.

### 2.8. Fluorescence In Situ Hybridization

Dual-color FISH analysis on paraffin sections of glioblastoma was employed to determine the status of the EGFR gene. The EGFR gene copies were measured using ZytoLight SPEC EGFR Green/CEN 7 Orange Dual Color Probe (ZytoVision, Bremerhaven, Germany) following the manufacturer’s instructions. In essence, section slides were deparaffinized, dried, and fixed in formaldehyde at a concentration of 4%. After that, each slide was probed in the designated area and covered with a plastic coverslip before being heated to accelerate both chromosomal and probe DNA denaturation. After hybridization in a humidified oven, the slices were washed and counterstained with DAPI. The slides were mounted with mounting media and inspected with a Zeiss Axiophot fluorescence microscope (Carl Zeiss). The presence of ≥15 copies of EGFR per cell in ≥10% of examined cells was considered as positive for EGFR gene amplification, according to the criteria [32].

### 2.9. Quantitative Real-Time Reverse Transcription-Quantitative Polymerase Chain Reaction

Total RNA was extracted from the GBM and normal brain tissues by using TRIzol™ reagents (Invitrogen; Thermo Fisher Scientific, Waltham, MA, USA). Reverse transcription-quantitative polymerase chain reaction (RT-PCR) was used to detect mRNA expression in GBM tissues. The primers used for this study are enumerated in Appendix A. All the selected mRNAs were predegenerated for 5 min at 95 °C and 1 min at 94 °C for 35 cycles. Next, they were predegenerated for 1 min at 56 °C, 2 min at 72 °C, and 10 min at 72 °C. The relative expression level of mRNA (gene) was calculated using 2 ^−∆∆Cq^ (2 ^−∆∆Cq^ ≥ 2 was regarded as high expression).

### 2.10. Cell Migration Assay

For the transwell migration assay, the cultured wild-type or shUSP6NL GBM cells were incubated for 24 h in 6-well plates (Costar, Washington, DC, USA) with RPMI and 10% FCS until they reached 95%–100% confluence. Chamber membranes (8 μm, BD Falcon) were not precoated with Matrigel before seeding with 1 × 10^5^ cells. RPMI with 2% FCS supplement was added to the upper chamber, and 600 μL of RPMI containing 20% FCS was added to the lower chamber. The cells were incubated for 48 h, with or without treatment. The noninvading cells on the top of membranes were carefully removed, and the invaded cells that penetrated the membrane were fixed in ethanol, followed by crystal violet staining. The number of invaded cells identified under the microscope in five random fields of vision were counted, and representative images were photographed.

### 2.11. Cell Invasion Assay

Cell invasiveness was evaluated using a Transwell Matrigel invasion assay (Costar). Chamber membranes (8 μm, BD Falcon) were not precoated with 6 µL Matrigel at 4 °C overnight before seeding with 2 × 10^4^ cells. RPMI with 2% FCS supplement was added to the upper chamber, and 600 µL of RPMI containing 20% FCS was added to the lower chamber. The cells were incubated for 48 h, with or without treatment. The noninvading cells on the top of the membranes were carefully removed, and the invaded cells that penetrated the membrane were fixed in ethanol, followed by crystal violet staining. The number of invaded cells was counted under the microscope in five random fields of vision, and representative images were photographed.

### 2.12. Coupled Cell Cycle and Cell Proliferation Assay

A 5′-bromo-2′-deoxyuridine (BrdU) flow kit (BD Pharmingen, San Diego, CA, USA) was used to determine the cell cycle kinetics and to measure the incorporation of BrdU into the DNA of proliferating cells. The assay was performed according to the manufacturer’s protocol. Briefly, T98G cells (2 × 10^5^ per well) were seeded overnight in 6-well tissue culture plates and treated with an optimized concentration of shRNAs in a medium containing 10% FBS for 72 h, followed by the addition of 10 μM BrdU, and incubation continued for an additional 30 min. Both the floating nonviable and adherent cells were pooled from triplicate wells per treatment point, fixed in a solution containing paraformaldehyde and the detergent, and then incubated for 1 h with DNase at 37 °C (30 μg per sample). FITC-conjugated anti-BrdU antibody (1:50 dilution in wash buffer; BD Pharmingen) was added and incubated for 20 min at room temperature. Cells were washed with wash buffer, and total DNA was stained with 7-aminoactinomycin D (7-AAD; 20 μL per sample), followed by a flow cytometric analysis of total DNA content (7-AAD) and cell cycle progression.

### 2.13. Apoptosis Assay

Induction of apoptosis was determined by using the commercial kit Annexin-V/PI for apoptosis assay. Following treatment of the both scramble control and USP6L knockdown cells (1 × 10^6^ cells/mL) with TMZ or control for 48 h, cells were harvested and washed twice with 2 mL of ice-cold phosphate-buffered saline (PBS). Afterwards, treated cells were incubated with 100 μL of HEPES buffer containing 2 μL of fluorescein isothiocyanate- (FITC-) conjugated annexin V and 2 μL of propidium iodide (PI) for 15 min. After washing the cells of excess reagents, 400 μL of binding buffer was added. The stained cells were immediately analyzed with a FACSCalibur flow cytometer. The experiment was performed in triplicate, and at least 10,000 counts were made for each sample. The percentage of apoptotic cells was calculated by CellQuest software (Becton, Dickinson and Co., San Jose, CA, USA).

### 2.14. Spheroid/Sphere Formation

The cells were seeded in serum-free low-adhesion culture plates containing RPMI1640 with B27 supplement (Invitrogen), 20 ng/mL EGF, and 20 ng/mL basic-FGF (stem cell medium; PeproTech, Rocky Hill, NJ, USA) for 2 weeks to allow tumorsphere formation. The spheres were counted under a microscope. The tumorsphere formation efficiency was calculated as the ratio of the number of tumorspheres formed to the seeded adherent cell number. For cell differentiation, 5 × 10^5^ single cells that dissociated from tumorspheres were cultured in a general 10-cm culture dish containing a complete medium for 2 weeks. Under this culture condition, the cells gradually lost their GFP expression, and the cells were harvested for further experiments.

### 2.15. Colony Formation Assay

Next, 2500 cells/cm^2^ were suspended in 0.3% agarose with MammoCult medium (StemCell Technologies, Vancouver, BC, Canada) and layered on a 0.8% agar base layer. The culture was covered with 0.5 mL of MammoCult medium and cultured for 14 days. For quantification, the wells were then imaged using a microscope, and the colonies were analyzed using ImageJ software.

### 2.16. Animal Studies

For in vivo studies, 1 × 10^6^ T98G-R cells suspended in 50 mL of Matrigel (BD Biosciences) were injected subcutaneously into the left flank of 5–6-week-old female NOD/SCID mice (10 mice per group). Animals were randomly divided into four treatment groups based on the type of cells with which the animals were inoculated: control, TMZ alone, shUSP6NL alone, and TMZ + shUSP6NL. Tumor growth was measured twice a week, and the volume was estimated as (length × width^2^)/2. Once tumors become incompatible with life, or at the end of the experiment on day 22, mice were humanely killed. The tumors were then extracted and analyzed. The animal study was approved by the Institutional Laboratory Animal Committee of Taipei Medical University (Approval number: LAC-2017-0512).

### 2.17. Statistical Analysis

All statistical analyses were performed using GraphPad Prism 6.0 (San Diego, CA, USA) and SPSS for Windows, version 25.0 (IBM, Armonk, NY, USA). The Pearson χ*^2^* test was used to assess the association between pairs of categorical variables. To indicate degree of agreement between two pathologists during grading of staining expression, Cohen’s Kappa was counted. Differences in continuous variables within the group were compared using paired Student *t* tests, one-way analysis of variance, and multivariate analysis. Kaplan–Meier curves were used to compare survival rates between groups. Statistical significance was set at *p* < 0.05.

## 3. Results

### 3.1. USP6NL and EGFR Overexpressed in Glioblastoma

To investigate the involvement of USP6NL and EGFR in human GBM, EGFR and USP6NL expression was detected in tumor and nontumor GBM tissue from patients with GBM. The RNA-seq expression data of the GBM and corresponding healthy control data were downloaded from the TCGA-GBM database (n = 370). Our results demonstrated that the mRNA expression of EGFR and USP6NL was higher in patients with GBM than in nontumor samples (Figure 1A,B). In addition, we observed a significant positive correlation between the expression of USP6NL and EGFR (R = 0.22, *p* < 0.05) in the TCGA-GBM cohort (Figure 1C). These results suggest a role for USP6NL and EGFR in the development of GBM and indicate a spatiotemporal and functional association between them. Basic clinical characteristics and the outcome of our in-house SHH-GBM cohort are described in Appendix A. To confirm the result of the TCGA-GBM cohort, we also observed expression of USP6NL in our SHH-GBM cohort, where the specimens were evaluated by relatively comparable pathologists (Cohen’s Kappa 0.78), as delineated in Appendix A. Our immunohistochemistry analysis of our own in-house SHH-GBM cohort (n = 60) revealed enhanced USP6NL expression, as represented by high Q-scores in the EGFR-amplified samples compared with their nonamplified or nontumor counterparts (nontumor << EGFR^amp−^ << EGFR^amp+^) (Figure 1D). Nontumor counterparts represented as a histologically normal part in adjacent area to the tumor portion (NAT). A relatively high Q-score in EGFR amplified tissue reflected and summarized both high positivity and intensity of USP6L expression in this subset, as opposed to non-EGFR amplified glioma tissue. A representative negative control and each grade of intensity of USP6NL expression in glioma tissue were also provided (Appendix A). Furthermore, the protein and mRNA expression of USP6NL in four GBM cell lines (U87MG, T98G, Hs683, and DBTRG05MG) were also assessed. Our data demonstrated that both protein and mRNA expression of USP6NL was enhanced in the U87MG and T98G cells compared with the Hs683 and DBTRG05MG cells (Figure 1E,F); therefore, for further experiments, we chose the U87MG and T98G cells. Taken together, these results suggest USP6NL as an interesting potential to later predict EGFR overexpression or amplification in gliomas that may subsequently have a different phenotype than other glioma tumors.

### 3.2. USP6NL and EGFR Positively Regulate TMZ Resistance in GBM Cells

To demonstrate the crucial role of USP6NL and EGFR in TMZ-resistant GBM cells, the GBM cells were exposed to low-dose TMZ for a long duration to establish a TMZ-resistant cell line (see Ujifuku et al. (2010) for a description of the method) [29]. Western blot analysis of USP6NL expression in GBM-TMZ-resistant cells demonstrated induced expression compared with parental cells (Figure 2A). The cell viability assay is shown in Figure 2B. U87MG-R and T98G-R cells were more resistant to TMZ compared with the parent U87MG and T98G cells. Furthermore, to demonstrate the role of USP6NL in GBM resistance, we knocked down the USP6NL in the shUSP6NL vector in both the GBM-TMZ-resistant cells (U87MG-R and T98G-R). The Western blot analysis results revealed significantly reduced USP6NL expression in the U87MG-R and T98G-R cells (Figure 2C) and notably, reduced EGFR expression (Figure 2C) in both the USP6NL-knockdown U87MG-R and T98G-R cells, demonstrating a positive association between USP6NL and EGFR in TMZ-resistant GBM cells. The proliferation of USP6NL-knockdown U87MG-R and T98G-R cells was examined using a CCK-8 assay. As shown in Figure 2D, under dose-dependent exposure of TMZ, loss of USP6NL reduced viability and change in morphology compared with the controls (scramble) in both the U87MG-R and T98G-R cells (Figure 2E). Enhanced TMZ-induced early and late apoptosis, cytotoxicity, and reduction in viable cells were observed in both the USP6NL-knockdown U87MG-R and T98G-R cells (Figure 2F).

### 3.3. Association of Drug Resistance with High USP6NL and Cancer Stemness Properties

Cancer stemness is associated with therapy resistance and the self-renewal properties of cancer cells [33]. We observed that both the U87MG-R and T98G-R cells demonstrated enhanced self-renewal (colony-forming ability), tumor-initiation (sphere-forming capacity), and invasive/migratory abilities compared with the parental control (Figure 3A–C). In addition, the protein and mRNA expression levels of USP6NL, pluripotency and cancer stem cell markers (CD44 and CD133), transcription factor (Nanog and SOX2), and efflux transporter ABCG2 were significantly overexpressed in the U87MG-R and T98G-R sphere compared with the parental control (Figure 3D,E). Similarly, U87MG-R and T98G-R demonstrated higher EMT (E-Cadherin and N-Cadherin) progress compared with the control (Figure 3F).

### 3.4. Abrogation of USP6NL Ameliorated the Stemness Phenotype of GBM Resistant Cells

The effect of USP6NL knockdown (shUSP6NL) on GBM-TMZ resistance was evaluated. As presented in Figure 4A, the self-renewal properties (colony-forming abilities) of U87MG-R and T98G-R cells were hampered when USP6NL was inhibited. In addition, the migratory, invasive, and tumor-initiating abilities (Figure 4B,C) of the cells were overtly reduced after USP6NL-knockdown in the U87MG-R and T98G-R cells compared with the control (scramble) transfected cells. Additionally, the Western blot analysis and qRT-PCR demonstrated that USP6NL abrogation in U87MG-R and T98G-R cells inhibits the expression of USP6NL, pluripotency and cancer stem cell markers (CD44 and CD133), transcription factors (Nanog and SOX2), and the efflux transporter (ABCG2), both at the protein and mRNA level in the U87MG-R and T98G-R (shUSP6NL transfected) cells compared with the control (scramble) cells (Figure 4D,E). Moreover, the expression of EMT markers was reduced after USP6NL-knockdown in both U87MG-R and T98G-R cells compared with the control cells (Figure 4F). This finding suggests that targeting and inhibiting USP6NL can lead to the reduction in tumor initiation, invasion, drug resistance, and disease relapse in GBM.

### 3.5. Effect of USP6NL on DNA Damage Repair Response

We evaluated how USP6NL modulates cancer drug resistance. The bioinformatics analysis of TCGA-GBM datasets demonstrated that the mRNA expression of USP6NL, EGFR (r^2^ = 0.177), RAD51 (r^2^ = 0.202), and BRCA2 (r^2^ = 0.282) was positively correlated (Figure 5A). RAD51 expression and the BRCA2 axis are new molecular targets for sensitizing glioma cells to alkylating anticancer drugs and are associated with DNA damage response (DDR) [34]. To understand the role of USP6NL in DDR, we used USP6NL-knockdown U87MG-R and T98G-R cells to examine yH2AX foci formation. yH2AX is a variant of the histone H2A family, and phosphorylation of H2Ax is a marker for regulating the repair of double-stranded DNA breaks [35]. As described in Figure 5B, USP6NL suppression resulted in reduced yH2AX foci, suggesting that USP6NL contributes to DNA damage repair. Furthermore, to examine the detailed role of USP6NL in DDR, the accumulation of RAD51 and BRCA2 together with yH2AX was assayed. As presented in Figure 5C, both the cells demonstrated USP6NL inhibition, resulting in a decrease in the number of RAD51/BRCA2 positive cells. This finding suggests that USP6NL mainly influences RAD51 and BRCA2 retention at the double-strand breaks (Figure 5D).

### 3.6. Inhibition of USP6NL Induces Autophagy of GBM Cells

We evaluated the coexpression of USP6NL and EGFR along with autophagy markers, such as ATG5, ATG7, Beclin-1, and LC3A/B, by using TCGA-GBM datasets. As presented in Figure 6A, the mRNA expression of USP6NL, EGFR was negatively correlated with the expression of Beclin-1 (r^2^ = −0.0421), ATG7 (r^2^ = −0.00932), and LC3A/B (r^2^ = 0.391) (Figure 6A). The immunofluorescence intensity of ATG5, ATG7, Beclin-1, and LC3A/B all tended to be induced in both the USP6NL-knockdown and TMZ (200 µM)-treated U87MG-R and T98G-R cells compared with the control scramble (Figure 6B); this result indicates the role of USP6NL in the evasion of cell death and is one of the hallmarks of cancer [36]. However, in the USP6NL-knockdown cells, U87MG-R and T98G-R induced the autophagic death of cells compared with the control, suggesting that USP6NL plays a key role in therapy resistance and evasion of cell death in GBM. Notably, as expected, the protein expression of the aforementioned autophagy marker was also induced in the presence of TMZ (200 µM) and the absence of USP6NL, as observed through Western blotting (Figure 6C). Furthermore, to demonstrate the interaction of USP6NL with EGFR and regulation of EGFR ubiquitination, we immunoprecipitated USP6NL (or EGFR) from the T98G cells, and then immunoblotted EGFR (or USP6NL) (Figure 6D,E). We observed that USP6NL demonstrated significant binding with EGFR within the GBM cell (T98G). Thus, we examined that the loss of USPNL markedly enhanced the ubiquitination activity of EGFR in GBM cells transfected with shUSP6NL compared with the control (NT) (Figure 6F), and the results indicated that USP6NL may play a vital role in the protection and accumulation of EGFR by promoting its deubiquitination and preventing its subsequent downregulation.

### 3.7. USP6NL Silencing Potentiates TMZ-Induced Cell Death In Vivo

The effect of USP6NL silencing was experimentally assessed in an in vivo animal model. Female nude mice bearing T98G-shUSP6NL and T98G-sh control cells were given in a vehicle of (0.5% DMSO in PBS) or TMZ (30 mg/kg) intraperitoneally on each side of the animals’ flanks. A significant reduction in the progression of shUSP6NL tumors was observed compared with the control (Figure 7A,B), even in the absence of TMZ treatment. Additionally, TMZ treatment considerably inhibited tumor growth in the shUSP6NL tumor compared with the control (Figure 7A,B). Tumor apoptosis was measured using the TUNEL reaction assay, and the result demonstrated that shUSP6NL significantly promoted apoptosis in tumor tissue of treated mice compared with the control group (Figure 7C).

## 4. Discussion

In this study, we explored the key molecular mechanisms underlying the development of malignant GBM, especially in connection with EGFR and the ubiquitination enzyme USP6NL. Many covalent modifications are involved in the early feedback loop that controls EGFR signaling in cancer progression [37], or the attenuation of EGFR signaling achieved through internalization or degradation of activated EGFR [38]. The inhibition of receptor internalization or degradation enhances EGFR-mediated signaling, which is crucial for the maintenance of cellular homeostasis. Many studies have identified the molecular workings of the key machinery executing EGFR ubiquitination and the key associated endocytic associated proteins [39].

Ubiquitin-specific proteases (USPs), which work as deubiquitinating enzymes (DUBs), are emerging as important regulators of cancer progression [40,41]. Abnormal USP family expression has been demonstrated in liver, oral, gastric, breast, and colorectal cancers [42]. USP4, USP7, and USP47 are DUBs that are abnormally overexpressed in many cancers [43,44]. USP6NL, a key USP, plays a key role in the regulation of Wnt/β-Catenin signaling in colorectal cancer [23]. However, USP6NL in association with EGFR in GBM has not been studied yet, especially in the context of GBM drug (TMZ) resistance. Drug resistance is induced by many internal factors, such as induced drug efflux and reduced drug uptake, as well as the alteration in cell cycle checkpoints, induced expression of emergency response genes, inhibition of apoptosis, autophagy, alteration of the drug target, and drug compartmentalization [16]. As shown in schematic Figure 8, this study provides the novel insights into the role of USP6NL/EGFR in GBM-TMZ resistance. We demonstrated that the overexpression of USP6NL was strongly correlated with EGFR in the GBM tumor tissue and cell lines. Furthermore, TMZ-resistant GBM cells were established, and an expression analysis demonstrated the induced expression of USP6NL and EGFR in TMZ-resistant cells compared with wild-type cells, indicating the key role of this duo in TMZ resistance. The cell cycle of cancer cells is significantly different from that of normal tissue cells, and the antiapoptotic property is also a hallmark of drug resistance [45]. Similarly, as shown in the apoptosis analysis, shRNA-mediated knockdown of TMZ-resistant GBM cells demonstrated induced apoptosis under the influence of TMZ, whereas control cells could withstand higher doses of TMZ, thus showing antiapoptotic action.

We demonstrated that TMZ-resistant GBM cells harbor the ability of self-renewal, by generating tumorspheres, in order to invade and enhance the EMT process. Following those phenotypes, it has been described that the expression of stem cell transcription factors, such as CD44, SOX2, CD133 ABCG2, and Nanog, play a pivotal role in determining the aggressiveness of cancer [46]. Therefore, we observed that these stemness markers, along with USP6NL, were induced in TMZ-resistant GBM cells. This suggests a strong correlation between USP6NL level and the self-renewal process. Furthermore, EMT-associated markers (N-cadherin upregulation of E-cadherin downregulation) were modulated in TMZ-resistant cells, confirming the EMT transition of GBM [46,47]. Furthermore, the shRNA-mediated knockdown of USP6NL resulted in reduced expression of the aforementioned markers, thus confirming the correlation between USP6NL and TMZ resistance.

We further demonstrated the relationship between histone H2AX phosphorylation and the sequestration of DNA repair factors upon the introduction of DSBs, the first cellular response is H2AX phosphorylation [48]. Colocalization of BRCA1-Y-H2AX-RAD51 is critically important as a DNA-binding factor during DSB [48]. Therefore, we evaluated BRCA1-Y-H2AX-RAD51 colocalization and observed a strong correlation between USP6NL and BRCA1-yH2AX-RAD51 in the TCGA-GBM datasets. Furthermore, expression of phosphorylated H2AX (y-H2AX) was reduced in the shRNA-mediated knockdown of USP6NL cells (shUSP6NL), and TMZ-resistant cells demonstrated higher expression and colocalization of y-H2AX-BRCA1-RAD51, suggesting that TMZ induced DNA repair. Furthermore, the expression of DNA repair factors was decreased in shUSP6NL GBM cells, highlighting the key role of the USP6NL axis in the DNA damage repair response.

Autophagy promotes both the survival and apoptotic death of GBM cells, and it can increase drug resistance in multiple cancers [49,50]; targeting autophagy is therefore an effective strategy for improving TMZ sensitivity in GBM [51,52]. Thus, denoting a specific target can help determine the therapeutic value of this complex physiological process. Among several well-known markers for autophagy, BECN-1, ATG7, and LC3A/B are known to represent autophagosome–lysosome fusion, the canonical autophagy pathway, and autophagic flux, respectively [53]. As expected, we noted that USP6NL and EGFR were strongly correlated with the autophagic markers (BECN-1, ATG7, and LC3A/B) in our in silico study. Furthermore, immunofluorescence and Western blot data demonstrated that shUSP6NL inhibition results in the reduction in expression markers associated with autophagy, whereas control and TMZ treatment reversed the effect caused by shUSP6NL inhibition in GBM cells, especially for the expression of autophagic markers. This suggests the importance of USP6NL and EGFR amplification for TMZ resistance in GBM.

Notably, the in vivo studies also demonstrated that the tumor growth rate and tumor size in the shUSP6NL + TMZ group were significantly lower than that in the control + TMZ treatment group. An increase in the apoptosis of the shUSP6NL group was also observed, providing strong evidence for the key role of USP6NL in targeting GBM-TMZ resistance and highlighting the possibility of subsequent human-targeted therapy research related to USP6NL.

## 5. Conclusions

This study provides novel insights into the USP6NL/EGFR axis that suppresses anticancer therapeutic responses, induces cancer invasiveness, and facilitates reduced sensitivity to TMZ treatment in GBM in an autolysosome-dependent manner. Therefore, controlling the USP6NL may offer an alternative, but efficient, therapeutic strategy for targeting and eradicating otherwise resistant and recurrent phenotypes of aggressive GBM cells.

## Figures and Tables

**Figure 1 biomedicines-10-01531-f001:**
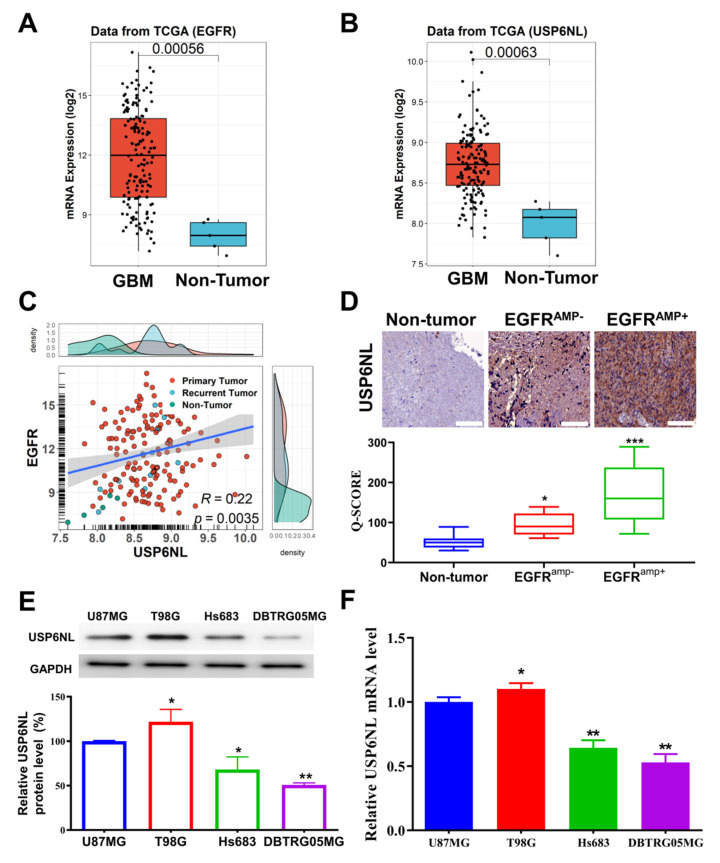
Expression analysis of USP6NL in EGFR^amp+/amp−^ in GBM samples. (**A**,**B**) mRNA level of EGFR and USP6NL in patients with GBM from The Cancer Genome Atlas (TCGA) dataset (n = 207) and corresponding healthy samples (n = 163). (**C**) Correlation analysis of mRNA expression of EGFR and USP6NL in the TCGA-GBM database. (**D**) Immunohistochemistry analysis of USP6NL expression in EGFR-amplified samples compared with their nonamplified SHH-GBM cohort (n = 60) (original magnification ×100). (**E**) Protein and (**F**) mRNA levels of USP6NL in four (U87MG, T98G, Hs683, and DBTRG05MG) GBM cell lines were assessed using Western blot and RT-PCR methods, respectively. *** *p* < 0.001, ** *p* < 0.01, * *p* < 0.05.

**Figure 2 biomedicines-10-01531-f002:**
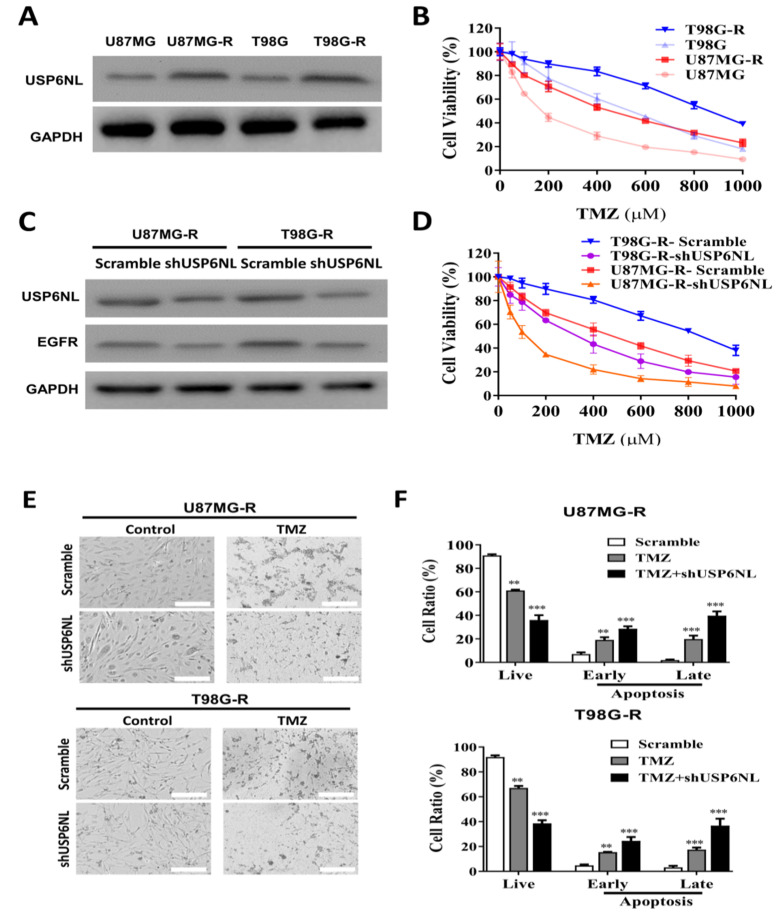
USP6NL is highly expressed in temozolomide (TMZ)-resistant GBM (U87MG-R and T98G-R) cells. (**A**) Representative image of Western blot analysis showing the USP6NL expression in TMZ-resistant GBM cells. (**B**) Viability assay performed using the CCK-8 assay kit showing that established TMZ-resistant cell lines, U87MG-R and U251-R, exhibited increased resilience toward TMZ treatment. (**C**) shRNA (shUSP6NL) knockdown of USP6NL. Western blot analysis demonstrated that the expression of USP6NL and EGFR was significantly reduced in the knockdown cells compared with the control. (**D**,**E**) Reduction in the cell viability and change in the morphology of TMZ-resistant GBM cells, demonstrating the effect of USP6NL-knockdown sensitized resistant cells toward TMZ. (**F**) Bar graph shows the representative proportion of apoptotic cells according to flow cytometry of Annexin V-FITC/PI staining in USP6NL-knockdown and control TMZ-resistant cells treated with TMZ. ** *p* < 0.01 and *** *p* < 0.001.

**Figure 3 biomedicines-10-01531-f003:**
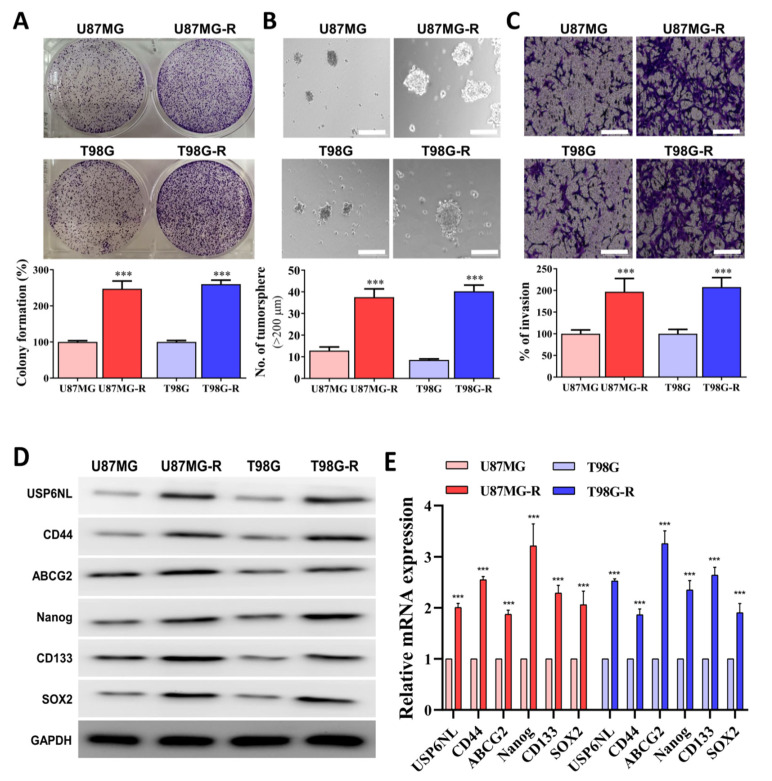
USP6NL is markedly expressed in TMZ resistant cells with high stemness phenotypes (**A**–**C**) Representative image of colony-forming (self-renewal ability), tumor initiation (sphere assay), and migratory/invasive properties of TMZ-resistant GBM cells (U87MG-R and T98G-R). Scale bar: 100 μm. (**D**,**E**) Western blot and qRT-PCR results of the expression of USP6NL and key CSC’s markers associated with therapy resistance. (**F**) EMT processes in TMZ-resistant GBM (U87MG-R and T98G-R) cells were also examined through IF analysis of E-cadherin and N-cadherin staining. Scale bar: 10 μm. *** *p* < 0.001.

**Figure 4 biomedicines-10-01531-f004:**
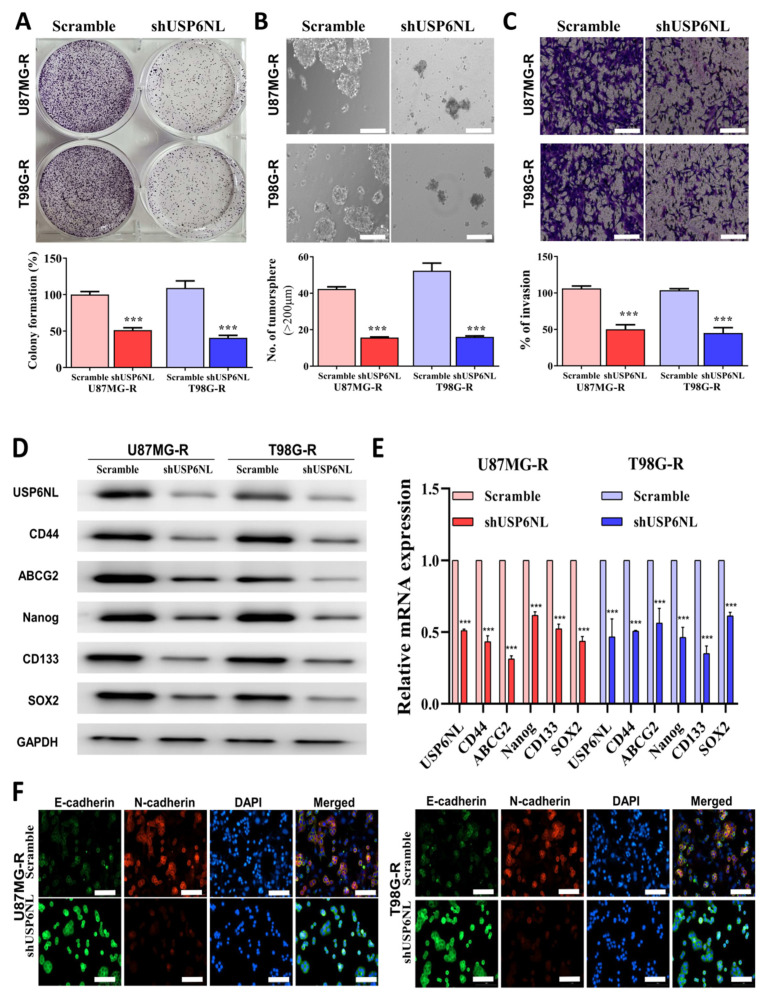
USP6NL inhibition regulates cancer stemness properties of GBM cells. (**A**–**C**) Representative image of colony-forming (self-renewal ability), tumor initiation (sphere assay), and migratory/invasive properties of USP6NL-knockdown GBM cells (U87MG-R and T98G-R). Scale bar: 100 μm. (**D**,**E**) Western blot and qRT-PCR results of the expression of key CSC’s markers associated with therapy resistance. (**F**) The EMT process in USP6NL-knockdown GBM (U87MG-R and T98G-R) cells was also examined through IF analysis of E-cadherin and N-cadherin staining. Scale bar: 10 μm. *** *p* < 0.001.

**Figure 5 biomedicines-10-01531-f005:**
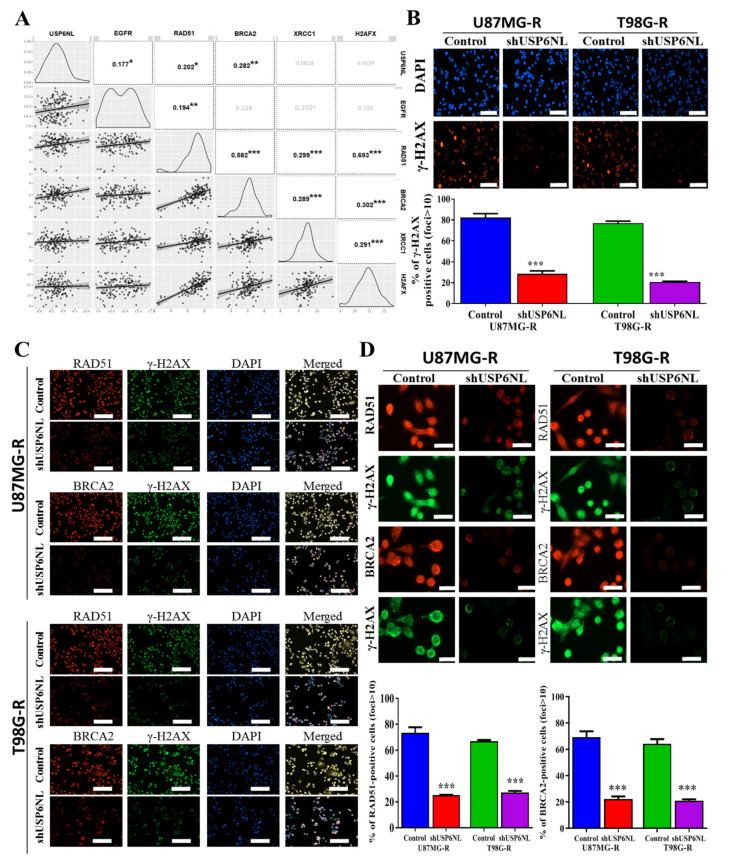
USP6NL regulates the DNA damage response. (**A**) Bioinformatics analysis of the correlation expression of USP6NL and EGFR (r^2^ = 0.177), with DNA repair response (DDR) markers RAD51 (r^2^ = 0.202) and BRCA2 (r^2^ = 0.282). (**B**,**C**) Immunostaining with indicated antibodies. The representative image shows that USP6NL depletion inhibits DNA damage repair by inhibiting the expression of RAD51 and BRCA2-positive cells; scale bar: 10 μm (**D**) Representative images of RAD51/BRCA2 foci in USP6NL-knockdown cells. * *p* < 0.05, ** *p* < 0.01 and *** *p* < 0.001.

**Figure 6 biomedicines-10-01531-f006:**
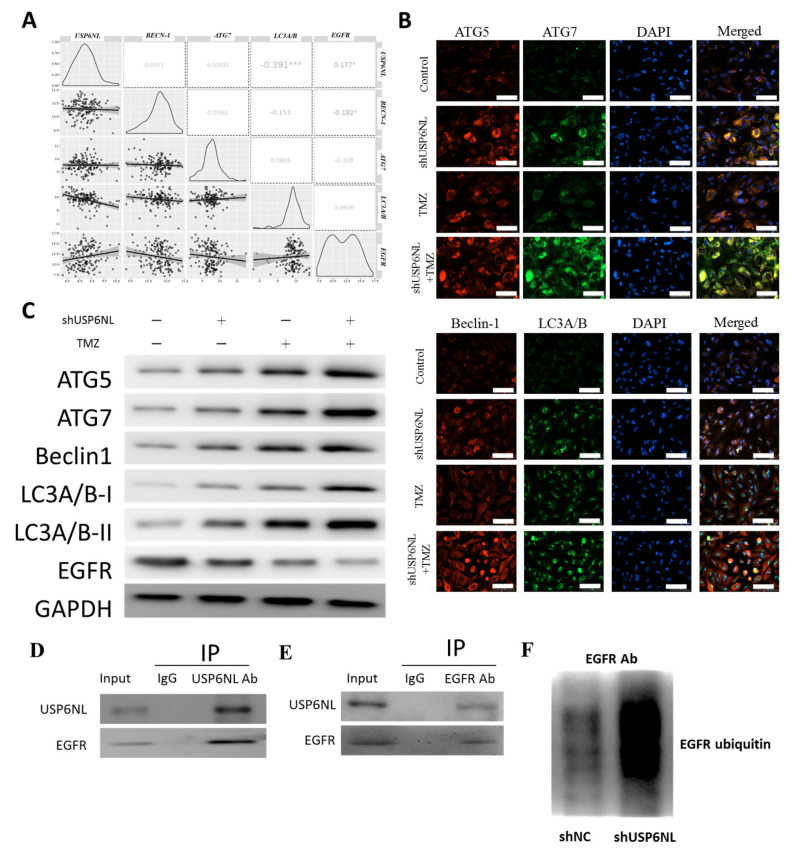
Effect of USP6NL inhibition on the expression of autophagy markers in GBM cells. (**A**) Bioinformatics analysis of the correlation expression of USP6NL and EGFR with autophagic markers ATG5, ATG7, Beclin-1, and LC3A/B. (**B**,**C**) Immunofluorescence image and Western blot analysis of the expression of autophagic markers after USP6NL inhibition and TMZ treatment. Scale bar: 10 μm (**D**–**F**) Coimmunoprecipitation to demonstrate the association of USP6NL with EGFR and the regulation of EGFR ubiquitination, with or without USP6NL. * *p* < 0.05, *** *p* < 0.001.

**Figure 7 biomedicines-10-01531-f007:**
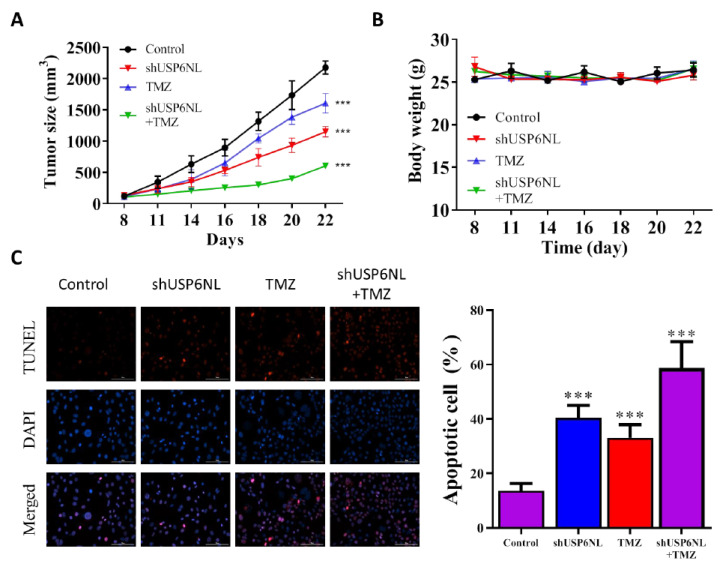
USP6NL-knockdown suppressed tumorigenicity of T98G GBM cells in nude mice. (**A**) Time course of control and shUSP6NL tumor volume progression, as measured using a caliper. (**B**) The effect of USP6NL-knockdown or TMZ treatment on the weights of tumor-bearing mice. (**C**) Apoptosis analysis in the tumor cells (day 22) by using the TUNEL assay (×200). *** *p*  <  0.001 vs. Control.

**Figure 8 biomedicines-10-01531-f008:**
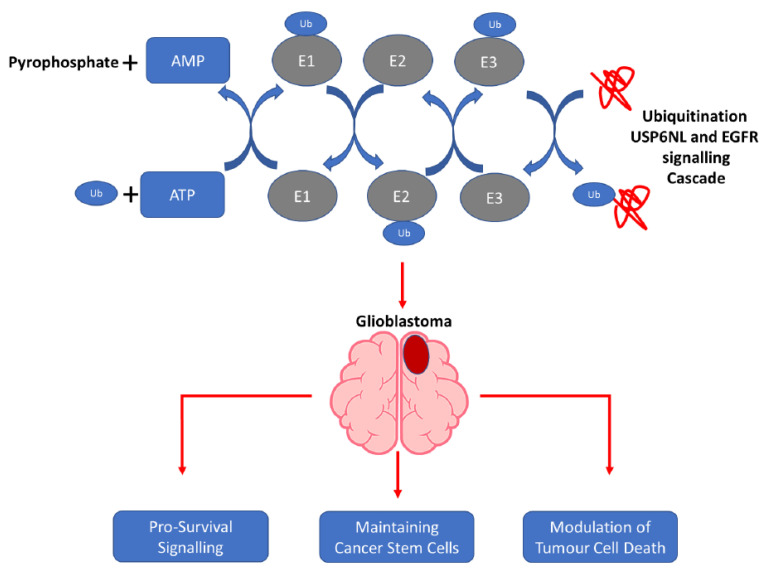
This study provides novel insights into the role of USP6NL/EGFR in GBM-TMZ resistance. The USP6NL/EGFR axis suppresses anticancer therapeutic responses, induces cancer invasiveness, and facilitates reduced sensitivity to TMZ treatment in patients with GBM in an autolysosome-dependent manner.

## Data Availability

The datasets used and analyzed in the current study are publicly accessible, as indicated in the manuscript.

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
