# Peer review of "Ubiquitin-Specific Protease 6 n-Terminal-like Protein (USP6NL) and the Epidermal Growth Factor Receptor (EGFR) Signaling Axis Regulates Ubiquitin-Mediated DNA Repair and Temozolomide-Resistance in Glioblastoma"

_biomedicines, 2022, doi:10.3390/biomedicines10071531_

Round 1

Reviewer 1 Report

Title

USP6NL and EGFR signaling axis regulates the ubiquitin-mediated DNA repair and temozolomide-resistance in glioblastoma.

Concise Summary

The authors aim that the ubiquitin-specific protease 6 N-terminal-like protein (USP6NL) affects GBM chemoresistance and tumorigenesis and that its inhibition may be a novel therapeutic strategy for GBM treatment. They propose that deubiquitinase USP6NL and growth factor receptor EGFR are strongly associated with the oncogenicity and resistance of GBM toward temozolomide. Methodologically, it is a combination of GBM cell culture, bioinformatic analysis of the TCGA-GBM database, shRNA transfection, western blotting, immunohistochemistry, quantitative real-time reverse transcription and test to evaluate different properties of tumor cells in culture. This methodology is combined with in vivo studies in mice to evaluate the tumor growth. It is stated that the elimination of USP6NL reversed the properties of GBM cells and resensitized them toward temozolomide by enhancing autophagy and reducing the DNA damage repair response. The conclude that controlling the USP6NL could contribute to the treatment of GBM.

Major criticisms

Title:

The acronyms USP6NL “ubiquitin-specific protease 6 N-terminal-like protein” and EGFR “epidermal growth factor receptor” should be translated in full.

Methodology:

1. Line 133. GBM cell culture. It is said that “….U251R cells were generated using a TMZ dose-escalation method.” However, there is no reference to this GBM cell line in results.

2. Immunohistochemistry evaluation. The methodology used to evaluate the staining positivity must be explained with more detail.

3. In addition, why did you decide >10% of immunostaining against USP6NL and EGFR as the reference cutoff value?

4. The authors state that “The staining positivity was detected using the percentage of positive cells and the intensity of staining.” Questions: Could you explain how you evaluate the intensity of staining. What immunohistochemical controls do you use?

5. The immunohistochemical procedure should be described with some more precision.

6. Line 171. The authors state that “All staining was reviewed and quantified independently by two pathologists blinded to the clinical outcome.”  This sentence is confusing. Questions: Do you know the clinical outcome of the patients included in the study? Incidentally, it had been interesting to include this data in the manuscript. In addition, which was the kappa coefficient of interobserver agreement?

7. Line 176. “The sections were blocked with serum for 30 min, incubated with primary antibodies …” Questions: Which antibodies were tested on the immunofluorescence study? In addition, the immunofluorescence procedure must be described with more detail.

8. Line 282. Fluorescent in situ hybridization (FISH) analysis of USP6NL expression has not been included in Methodology.

Results:

9. Lines 269-71. It is said that “Consistent with this finding, fluorescent in situ hybridization analysis of our own in-house SHH-GBM cohort (n = 60) revealed enhanced USP6NL expression in the EGFR-amplified samples compared with their nonamplified or nontumor counterparts (nontumor << EGFRamp- << EGFRamp+).” However, in Figure 1D, lines 282-283. this figure corresponds to immunohistochemical results of USP6NL.

10. Even more, there are no scale bars in the figure 1D. It would be helpful to have larger fields of view to show the quality of the immunohistochemistry.

11. Figure 1D.  In addition, the immunohistochemical evaluation of USP6NL and EGFR (both percentage of positive cells and intensity of staining) are not represented on the text.

12. Figure 1D. An explanation about  “Q-Score” should be added to Methodology Section.

Discussion:

13. Line 475-476. References should not be included in Discussion without adequate previous explanation. For example, “we noted that USP6NL and EGFR were strongly correlated with the autophagic markers (BECN-1, ATG7, and LC3A/B [49]) in our in silico study”. This text is confusing by two reasons: 1. It could be interpreted that the authors have produced the referenced article. 2. This reference is included without previous explanation  because this article is brought up.

14. Lines 443-461. In addition, a new reference of figures /figur1, figure 2, and so on) should be omitted in Discussion.

Conclusion:

15. Figure 8 cannot be included in this Section. The authors should limit themselves to briefly indicate the final message.

Minor criticisms

1. Line 288. The acronym USL6NL is misspelled.

2. Line 442. The authors state that. “We demonstrated that overexpression of USP6NL was strongly correlated with EGFR in the patient.” This sentence has no sense: It is not in the patient, but in the tumor.

3. Line 462. It is said that “We further demonstrated the relationship between histone H2AX phosphorylation and the sequestration of DNA repair factors [44].” It is confusing because it could be interpreted that the referenced article was produced by the authors of the present manuscript, but it is not (Gottlieb, R.A.; Andres, A.M.; Sin, J.; Taylor, D.P.J. Untangling autophagy measurements: All fluxed up. Circ Res 2015, 116, 624).

4. Lines 453-55. The same about “Expression of stem cell transcription factors [42], CD44, SOX2, CD133 ABCG2, and NONG, together with USP6NL, was induced in TMZ-resistant GBM cells. This suggests a strong correlation between USP6NL level and self-renewal process (Figure 3).” This sentence is confusing. What is the reason to quote this article? (Bao, B.; Ahmad, A.; Azmi, A.S.; Ali, S.; Sarkar, F.H. Overview of cancer stem cells (cscs) and mechanisms of their regulation: Implications for cancer therapy. Curr Protoc Pharmacol 2013, Chapter 14, Unit-14.25.)

5. Line 467. It is said that “Furthermore, y-H2AX expression was reduced in shUSP6NL cells”. In the case of abbreviations, please indicate what they are stand for.

Conclusion

Finally, I consider that it is an interesting and original work that gives relevant information about a possible effect of USP6NL on GBM cells and that it could contribute to a better knowledge of the role of this molecule in the oncogenesis and treatment of GBM. However, the authors should modify the text according to the reviewer considerations.

Author Response

Point-by-point responses to Editor’s comments:

We thank Editor for carefully reading our manuscript and providing valuable comments. We believe making use of all these comments has further helped improve the quality and appeal of our work, as well as strengthened the manuscript. Below are our point-by-point responses.

# Comments from Reviewer 1

Title:

The acronyms USP6NL “ubiquitin-specific protease 6 N-terminal-like protein” and EGFR “epidermal growth factor receptor” should be translated in full.

Answer: We thank the editor for the insightful suggestion. Indeed, the full translation of the acronym will further improve clarity of our interest axis. Thus, we have since updated our title.

Updated Title: Ubiquitin-specific protease 6 n-terminal-like protein (USP6NL) and epidermal growth factor receptor (EGFR) signaling axis regulates the ubiquitin-mediated DNA repair and temozolomide-resistance in glioblastoma

Methodology:

  1. Line 133. GBM cell culture. It is said that “….U251R cells were generated using a TMZ dose-escalation method.” However, there is no reference to this GBM cell line in results.

Answer: We appreciate the editor's insightful suggestion. We appeared to have misplaced this vital information. Therefore, we have added the reference in response to your comment [1,2]. Please kindly refer to line 128-136 and updated reference section.

2.1. GBM cell culture

Hs683, T98G, DBTRG05MG, and U87MG GBM cell lines were obtained from the Ameri-can Form Culture Collection (ATCC; Manassas, VA, USA) and maintained in an incubator with 5% CO2 in humidified air. The cells were cultured in Dulbecco's modified Eagle’s medi-um (#12491023; GIBCO, Life Technologies, Carlsbad, CA, USA) supplemented with 10% fetal bovine serum (GIBCO, Life Technologies), penicillin (100 IU/mL), and streptomycin (100 g/mL) (#15140122, GIBCO, Life Technologies). TMZ-resistant U87MGR and U251R cells were generated using a TMZ dose-escalation method up to 150 µM and then maintained at 100 µM TMZ for in vitro and in vivo experiments according to earlier reference [29,30].

New Reference:

  1. Ujifuku, K.; Mitsutake, N.; Takakura, S.; Matsuse, M.; Saenko, V.; Suzuki, K.; Hayashi, K.; Matsuo, T.; Kamada, K.; Nagata, I.J.C.l. MiR-195, miR-455-3p and miR-10a∗ are implicated in acquired temozolomide resistance in glioblastoma multiforme cells. 2010, 296, 241-248.
  2. Zhang, J.; Stevens, M.F.G.; Laughton, C.A.; Madhusudan, S.; Bradshaw, T.D. Acquired Resistance to Temozolomide in Glioma Cell Lines: Molecular Mechanisms and Potential Translational Applications. Oncology 2010, 78, 103-114, doi:10.1159/000306139.

  1. Immunohistochemistry evaluation. The methodology used to evaluate the staining positivity must be explained with more detail.

Answer: We appreciate the thoughtful recommendation of the reviewer. This crucial knowledge looked to have been undermined. In response to your criticism, we rearranged a new paragraph to provide more specific information about our immunohistochemistry evaluation. Please kindly refer to our updated material and methods section, specifically subsection of Immunohistochemistry staining in line 167 to 183.

2.6. Immunohistochemistry staining

Immunohistochemistry staining was performed on formalin-fixed paraffin-embedded sections by using antibodies against USP6NL. Briefly, tissue sections (4 µm) were made and deparaffinized. Following antigen retrieval, blocking solution was applied, then slices were incubated with primary antibodies against USP6NL (Creative Diagnostics Cat# DCABH-17363, RRID: AB_2489260) with dilution ratio of 1:100 for 2 h at room temperature. After in-cubation with Horseradish peroxidase (HRP) and Diaminobenzidine (DAB) chromogen along with substrate, slides were counterstained with hematoxylin, and final mounting solution was applied. As a negative control, a comparable staining method was employed using iso-type rabbit IgG for primary antibody. Cohen’s Kappa values for each pathologist, assessing expression quick score of USP6L in glioma tissue in Supplementary Table S3. Two inde-pendent pathologists assessed and scored USP6NL expression using the quick-score (Q-score), which is derived on staining intensity (I) and the proportion of stained cells (P) as previously described [29]. There were four different levels of staining intensity: 0 (no staining), 1+ (weak), 2+ (moderate), and 3+ (strong). Therefore, we finally counted the score as Q = I × P, with a max-imum score of 300. Representative staining of USP6L and control in glioma FFPE tissue sec-tions in Supplementary Figure S1.

  1. In addition, why did you decide >10% of immunostaining against USP6NL and EGFR as the reference cutoff value?

Answer: Thank you for the helpful suggestions from the reviewers. It was possible that we ignored something crucial when writing detailed information regarding our previous staining technique that created a substantial misunderstood. In this study, we actually did not separate the results of IHC staining of USP6NL into qualitative categories such as positive and negative, instead we were focusing on the quantification of quick-score to differentiate degree of expression between normal brain tissue and GBM tissue, then separated GBM into EGFR amplified and non-amplified based on FISH technique. However, in the context of EGFR amplification, by using FISH technique, based on most criteria [3], EGFR amplification was defined when the amplified EGFR signal were detected in ≥10% of analyzed cells. Therefore, we reconstructed this information into a new subsection based on our actual condition previously. Please kindly refer to our updated material and methods section, specifically subsection of Immunohistochemistry staining and subsection of FISH staining in line 201 ~ 211.

2.8. Fluorescence In Situ Hybridization

Dual-color FISH analysis on paraffin sections of glioblastoma was employed to deter-mine the status of the EGFR gene. The EGFR gene copies were measured using ZytoLight SPEC EGFR Green/CEN 7 Orange Dual Color Probe (ZytoVision, Bremerhaven, Germany) fol-lowing the manufacturer's instructions. In an essence, section slides were deparaffinized, dried, and fixed in formaldehyde at a concentration of 4%. After that, each slide was probed in the designated area and covered with a plastic coverslip before being heated to accelerate both chromosomal and probe DNA denaturation. After hybridization in a humidified oven, the slices were washed and counterstained with DAPI. The slides were mounted with mounting media and inspected with a Zeiss Axiophot fluorescence microscope (Carl Zeiss). The pres-ence of ≥15 copies of EGFR per cells in ≥ 10% of examined cells was considered as positive for EGFR gene amplification according to the criteria [30].

  1. The authors state that “The staining positivity was detected using the percentage of positive cells and the intensity of staining.” Questions: Could you explain how you evaluate the intensity of staining. What immunohistochemical controls do you use?

Answer: We appreciate the thoughtful recommendation of the reviewer. In response to your criticism, we then added the crucial information regarding assessment of intensity including grading of intensity and negative control into a new rearranged paragraph. Here, we used isotype antibody for our representative negative control. Furthermore, we also added new supplementary figure to provide the staining result of negative control and each grading of intensity. Please kindly refer to our updated material and methods section, specifically subsection of Immunohistochemistry staining and the Supplementary Figure S1.

Figure S1. Representative staining of USP6L and control in glioma FFPE tissue sections. Panel (A) is demonstrating a representative picture of negative control (glioma tissue stained with isotype rabbit IgG antibody), panel (B) is a representative picture of a 1+ positive glioma sample, panel (C) a representative picture of a 2+ positive glioma sample, and panel (D) a representative picture of a 3+ positive or the highest intensity of USP6L expression in glioma sample. Scale bar = 100 µm.

  1. The immunohistochemical procedure should be described with some more precision.

Answer: We greatly appreciate the attentive suggestion of the reviewer. In response to that advice, we rearranged a new paragraph to provide more specific information such as summary of step to stain the specimen, primary antibody, dilution ratio, and evaluation of staining expression. Please kindly refer to our updated material and methods section, specifically subsection of Immunohistochemistry staining in line 167 to 183.

2.6. Immunohistochemistry staining

Immunohistochemistry staining was performed on formalin-fixed paraffin-embedded sections by using antibodies against USP6NL. Briefly, tissue sections (4 µm) were made and deparaffinized. Following antigen retrieval, blocking solution was applied, then slices were incubated with primary antibodies against USP6NL (Creative Diagnostics Cat# DCABH-17363, RRID: AB_2489260) with dilution ratio of 1:100 for 2 h at room temperature. After in-cubation with Horseradish peroxidase (HRP) and Diaminobenzidine (DAB) chromogen along with substrate, slides were counterstained with hematoxylin, and final mounting solution was applied. As a negative control, a comparable staining method was employed using iso-type rabbit IgG for primary antibody. Cohen’s Kappa values for each pathologist, assessing expression quick score of USP6L in glioma tissue in Supplementary Table S3. Two inde-pendent pathologists assessed and scored USP6NL expression using the quick-score (Q-score), which is derived on staining intensity (I) and the proportion of stained cells (P) as previously described [29]. There were four different levels of staining intensity: 0 (no staining), 1+ (weak), 2+ (moderate), and 3+ (strong). Therefore, we finally counted the score as Q = I × P, with a max-imum score of 300. Representative staining of USP6L and control in glioma FFPE tissue sec-tions in Supplementary Figure S1.

  1. 6. Line 171. The authors state that “All staining was reviewed and quantified independently by two pathologists blinded to the clinical outcome.”  This sentence is confusing. Questions:Do you know the clinical outcome of the patients included in the study? Incidentally, it had been interesting to include this data in the manuscript. In addition, which was the kappa coefficient of interobserver agreement?

Answer: We impressively appreciate the constructive suggestion of the reviewer. In response to that question, we further add supplementary table to describe information about clinical characteristics and outcome of our in-house GBM cohort that we obtained for the specimens. We also assess the inter-observer agreement of our previously pathologists and determine relatively strong kappa coefficient (0.78, CI 95% 0.63 to 0.93). Therefore, we provided supplementary table for the result of inter-rater observer agreement between our two pathologists during the grading assessment of USP6L expression (0 to +3 grade of intensity) according to a total of 20 specimens consisting of normal and GBM tissues. The calculation of Cohen’s kappa is also added following method of statistical analysis. Please kindly refer to our updated material and methods section, specifically subsection of Immunohistochemistry staining in line 167 to 183.

Supplementary Table S3. Cohen’s Kappa values for each pathologist, assessing expression Q-score of USP6L in glioma tissue.

Pathologist_A

Pathologist_B

0-75

76-150

151-225

226-300

0-75

2

0

0

0

2 (6,7%)

76-150

1

8

0

0

9 (30,0%)

151-225

0

5

5

1

11 (36,7%)

226-300

0

0

0

8

8 (26,7%)

3
(10,0%)

13
(43,3%)

5
(16,7%)

9
(30,0%)

30

Weighted Kappaa

                                              0,78170

Standard error

0,07644

95% CI

0,63188 to 0,93153

P-value

1.07e-09

                        a Linear weights

Supplementary Table S1. Baseline clinical characteristics and outcome of GBM patients in SHH-GBM cohort

Characteristics (n=60)

Cases

Percentage (%)

Age

Less than or equal 60 y.o

34

52.3%

More than 60 y.o

31

47.7%

Gender

Male

32

49.2%

Female

33

50.8%

Tumor Location

Frontal

17

26.2%

Parietal

6

9.2%

Temporal

21

32.3%

Occipital

1

1.5%

Others

20

30.8%

Recurrences

Yes

3

4.6%

No

62

95.4%

y.o: years old

  1. Line 176. “The sections were blocked with serum for 30 min, incubated with primary antibodies …”Questions: Which antibodies were tested on the immunofluorescence study? In addition, the immunofluorescence procedure must be described with more detail.

Answer: We gratefully appreciate the constructive recommendation of the reviewer. In response to that advice, we rearranged a new paragraph to provide more specific information about our immunofluorescence procedure. Please kindly refer to our updated material and methods section, specifically subsection of Immunofluorescence staining in line 184 to 199.

2.7. Immunofluorescence staining

Representative human glioblastoma cell lines harboring either scramble control or knockdown of USP6L (shUSP6L) were plated in six-well chamber slides for 24 h to perform immunofluorescence analysis. The cells were fixed with 2% paraformaldehyde and probed with primary antibodies. To determine the positive signal, a fluorophore-conjugated second-ary antibody was added following examination using a Zeiss Axiophot (Carl Zeiss) fluores-cence microscope. The nuclei of viable cells were detected through 4′,6-diamidino-2-phenylindole (DAPI) staining. The primary antibodies being used were purchased from sev-eral different companies and diluted under specific concentrations, such as E-cadherin (1:100, #3195, rabbit mAb; Cell Signaling), N-cadherin (1:100, #13116, rabbit mAb; Cell Signaling), yH2AX (1:100, #7631, rabbit mAb; Cell Signaling), RAD51 (1:100, ab63801, rabbit polyclonal-Ab; Abcam), BRCA2 (1:100, # PA5-96128, rabbit polyclonal-Ab; ThermoFisher), ATG5 (1:100, ab228668, rabbit polyclonal-Ab; Abcam), ATG7 (1:100, ab133528, rabbit mAb; Abcam), Beclin-1 (1:100, ab62557, rabbit polyclonal-Ab; Abcam), and LC3A/B (1:100, #4108, rabbit polyclonal-Ab; Cell Signaling). The negative controls were carried out by omitting the primary antibody.

  1. Line 282. Fluorescent in situhybridization (FISH) analysis of USP6NL expression has not been included in Methodology.

Answer: It was a pleasure to receive very constructive feedback from the reviewer. In response to that advice, we added a new paragraph to provide more specific information about the FISH staining analysis of EGFR. Furthermore, we have already updated the defining criteria of EGFR FISH positive staining according to previous report that we cited into the new reference [3].  Please kindly refer to our updated material and methods section, specifically subsection of Fluorescence In Situ Hybridization and updated reference section in line 200-211.

2.8. Fluorescence In Situ Hybridization

Dual-color FISH analysis on paraffin sections of glioblastoma was employed to deter-mine the status of the EGFR gene. The EGFR gene copies were measured using ZytoLight SPEC EGFR Green/CEN 7 Orange Dual Color Probe (ZytoVision, Bremerhaven, Germany) fol-lowing the manufacturer's instructions. In an essence, section slides were deparaffinized, dried, and fixed in formaldehyde at a concentration of 4%. After that, each slide was probed in the designated area and covered with a plastic coverslip before being heated to accelerate both chromosomal and probe DNA denaturation. After hybridization in a humidified oven, the slices were washed and counterstained with DAPI. The slides were mounted with mounting media and inspected with a Zeiss Axiophot fluorescence microscope (Carl Zeiss). The pres-ence of ≥15 copies of EGFR per cells in ≥ 10% of examined cells was considered as positive for EGFR gene amplification according to the criteria [30].

New Reference:

  1. Lopez-Gines, C.; Gil-Benso, R.; Ferrer-Luna, R.; Benito, R.; Serna, E.; Gonzalez-Darder, J.; Quilis, V.; Monleon, D.; Celda, B.; Cerdá-Nicolas, M. New pattern of EGFR amplification in glioblastoma and the relationship of gene copy number with gene expression profile. Modern Pathology 2010, 23, 856-865, doi:10.1038/modpathol.2010.62.

Results:

  1. Lines 269-71. It is said that “Consistent with this finding, fluorescent in situ hybridization analysis of our own in-house SHH-GBM cohort (n = 60) revealed enhanced USP6NL expression in the EGFR-amplified samples compared with their nonamplified or nontumor counterparts (nontumor << EGFRamp- << EGFRamp+).” However, in Figure 1D, lines 282-283. this figure corresponds to immunohistochemical results of USP6NL.

Answer: We thank the reviewer for their input. Indeed, we incorrectly wrote for this information that actually figure 1D depicted the immunohistochemistry staining result of USP6NL. Therefore, we have since updated the sentence in response to your comment. Please kindly refer to line 303-324 and updated figure legend of Fig. 1D.

3.1. USP6NL and EGFR overexpressed in glioblastoma

To investigate the involvement of USP6NL and EGFR in human GBM, EGFR and USP6NL expression was detected in tumor and nontumor GBM tissue from patients with GBM. The RNA-seq expression data of the GBM and corresponding healthy control data were downloaded from the TCGA-GBM database (n = 370). Our results demonstrated that the mRNA expression of EGFR and USP6NL was higher in patients with GBM than in nontumor samples (Figure 1 A, B). In addition, we observed a significant positive correlation between the expression of USP6NL and EGFR (R = 0.28, p < 0.05) in the TCGA-GBM cohort (Figure 1C). These results suggest a role for USP6NL and EGFR in the development of GBM and indicate a spatiotemporal and functional association between them. Basic clinical characteristics and outcome of our in-house SHH-GBM cohort were described in Supplementary Table S1. To confirm the result of TCGA-GBM cohort, we also observed expression of USP6NL in our SHH-GBM cohort whom the specimens were evaluated by relatively comparable pathologists (Co-hen’s kappa 0.78) as delineated in Supplementary Table S3. Our immunohistochemistry analysis of our own in-house SHH-GBM cohort (n = 60) revealed enhanced USP6NL expres-sion as represented by high Q-score in the EGFR-amplified samples compared with their nonamplified or nontumor counterparts (nontumor << EGFRamp- << EGFRamp+) (Figure 1D). A relatively high Q-score in EGFR amplified tissue reflected and summarized both high positivi-ty and intensity of USP6L expression in this subset as opposed to non-EGFR amplified glioma tissue. Representative negative control and each grade of intensity of USP6NL expression in glioma tissue were also provided (Supplementary Figure S1). Furthermore, the protein and mRNA expression of USP6NL in four GBM cell lines (U87MG, T98G, Hs683, and DBTRG05MG) were also assessed. Our data demonstrated that both protein and mRNA ex-pression of USP6NL was enhanced in the U87MG and T98G cells compared with the Hs683 and DBTRG05MG cells (Figure 1E, F); therefore, for further experiments, we chose U87MG and T98G cells. Taken together, these results suggest USP6NL as an interesting prognostic marker in gliomas.

Figure 1. Expression analysis of USP6NL in EGFRamp+/amp− in GBM samples. (A, B) mRNA level of EGFR and USP6NL in patients with GBM from The Cancer Genome Atlas (TCGA) data set (n = 207) and corresponding healthy samples (n = 163). (C) Correlation analysis of mRNA expression of EGFR and USP6NL in the TCGA-GBM database. (D) Immunohistochemistry analysis of USP6NL expression in EGFR-amplified samples compared with their nonamplified SHH-GBM cohort (n = 60) (original magnification ×100). (E) Protein and (F) mRNA levels of USP6NL in four (U87MG, T98G, Hs683, and DBTRG05MG) GBM cell lines were assessed using Western blot and RT-PCR methods, respectively. ***P < 0.001, **P < 0.01,*P < 0.05.

  1. Even more, there are no scale bars in the figure 1D. It would be helpful to have larger fields of view to show the quality of the immunohistochemistry.

Answer: It was a pleasure to receive excellent comment from the reviewer. Accordingly, we have updated the new clearer resolution featured with scale bars in figure 1D. Please kindly refer to our new figure 1D.

  1. Figure 1D.  In addition, the immunohistochemical evaluation of USP6NL and EGFR (both percentage of positive cells and intensity of staining) are not represented on the text.

Answer: We appreciate for receiving a very positive comment from the reviewer. In our immunohistochemistry staining, we used namely Q-score that was based on multiplication of percentage of positivity and intensity of staining. Thus, this score already represented both staining parameters. However, to make a clear statement regarding the result of both features, we added new sentence and information in this paragraph. Therefore, please kindly refer to line 303-324.

3.1. USP6NL and EGFR overexpressed in glioblastoma

To investigate the involvement of USP6NL and EGFR in human GBM, EGFR and USP6NL expression was detected in tumor and nontumor GBM tissue from patients with GBM. The RNA-seq expression data of the GBM and corresponding healthy control data were downloaded from the TCGA-GBM database (n = 370). Our results demonstrated that the mRNA expression of EGFR and USP6NL was higher in patients with GBM than in nontumor samples (Figure 1 A, B). In addition, we observed a significant positive correlation between the expression of USP6NL and EGFR (R = 0.28, p < 0.05) in the TCGA-GBM cohort (Figure 1C). These results suggest a role for USP6NL and EGFR in the development of GBM and indicate a spatiotemporal and functional association between them. Basic clinical characteristics and outcome of our in-house SHH-GBM cohort were described in Supplementary Table S1. To confirm the result of TCGA-GBM cohort, we also observed expression of USP6NL in our SHH-GBM cohort whom the specimens were evaluated by relatively comparable pathologists (Co-hen’s kappa 0.78) as delineated in Supplementary Table S3. Our immunohistochemistry analysis of our own in-house SHH-GBM cohort (n = 60) revealed enhanced USP6NL expres-sion as represented by high Q-score in the EGFR-amplified samples compared with their nonamplified or nontumor counterparts (nontumor << EGFRamp- << EGFRamp+) (Figure 1D). A relatively high Q-score in EGFR amplified tissue reflected and summarized both high positivi-ty and intensity of USP6L expression in this subset as opposed to non-EGFR amplified glioma tissue. Representative negative control and each grade of intensity of USP6NL expression in glioma tissue were also provided (Supplementary Figure S1). Furthermore, the protein and mRNA expression of USP6NL in four GBM cell lines (U87MG, T98G, Hs683, and DBTRG05MG) were also assessed. Our data demonstrated that both protein and mRNA ex-pression of USP6NL was enhanced in the U87MG and T98G cells compared with the Hs683 and DBTRG05MG cells (Figure 1E, F); therefore, for further experiments, we chose U87MG and T98G cells. Taken together, these results suggest USP6NL as an interesting prognostic marker in gliomas.

  1. Figure 1D. An explanation about “Q-Score” should be added to Methodology Section.

Answer: We appreciate for receiving a very constructive input from the reviewer. According to this comment, we have updated the explanation of Q-score calculation in the methodology section according to previous report that we added this reference into the new citation [4]. Therefore, please kindly refer to line 174-178 and our additional new reference in reference section.

2.6. Immunohistochemistry staining

Immunohistochemistry staining was performed on formalin-fixed paraffin-embedded sections by using antibodies against USP6NL. Briefly, tissue sections (4 µm) were made and deparaffinized. Following antigen retrieval, blocking solution was applied, then slices were incubated with primary antibodies against USP6NL (Creative Diagnostics Cat# DCABH-17363, RRID: AB_2489260) with dilution ratio of 1:100 for 2 h at room temperature. After in-cubation with Horseradish peroxidase (HRP) and Diaminobenzidine (DAB) chromogen along with substrate, slides were counterstained with hematoxylin, and final mounting solution was applied. As a negative control, a comparable staining method was employed using iso-type rabbit IgG for primary antibody. Cohen’s Kappa values for each pathologist, assessing expression quick score of USP6L in glioma tissue in Supplementary Table S3. Two inde-pendent pathologists assessed and scored USP6NL expression using the quick-score (Q-score), which is derived on staining intensity (I) and the proportion of stained cells (P) as previously described [29]. There were four different levels of staining intensity: 0 (no staining), 1+ (weak), 2+ (moderate), and 3+ (strong). Therefore, we finally counted the score as Q = I × P, with a max-imum score of 300. Representative staining of USP6L and control in glioma FFPE tissue sec-tions in Supplementary Figure S1.

New reference:

  1. Detre, S.; Saclani Jotti, G.; Dowsett, M. A "quickscore" method for immunohistochemical semiquantitation: validation for oestrogen receptor in breast carcinomas. Journal of Clinical Pathology 1995, 48, 876-878, doi:10.1136/jcp.48.9.876.

Discussion:

  1. Line 475-476. References should not be included in Discussion without adequate previous explanation. For example, “we noted that USP6NL and EGFR were strongly correlated with the autophagic markers (BECN-1, ATG7, and LC3A/B [49]) in our in silico study”. This text is confusing by two reasons: 1. It could be interpreted that the authors have produced the referenced article. 2. This reference is included without previous explanation because this article is brought up.

Answer: We thank the editor for the insightful suggestion. Based on the comments previously, we think that situation is more closely related to the second reason, that the reference was inserted without further explanation of why cited this reference. Therefore, we added more elaboration about this reference into the following sentence for further integration. Please kindly refer to our discussion, particularly in line 525-529.

Autophagy promotes both the survival and apoptotic death of GBM cells, and it can in-crease drug resistance in multiple cancers [47,48]; targeting autophagy is therefore an effective strategy for improving TMZ sensitivity in GBM [49,50]. Thus, denoting a specific target can help determine the therapeutic value of this complex physiological process. Among several well-known markers for autophagy, BECN-1, ATG7, and LC3A/B are known to represent au-tophagosome–lysosome fusion, canonical autophagy pathway, and autophagic flux, respec-tively [51]. As expected, we noted that USP6NL and EGFR were strongly correlated with the autophagic markers (BECN-1, ATG7, and LC3A/B) in our in silico study.

  1. Lines 443-461. In addition, a new reference of figures /figur1, figure 2, and so on) should be omitted in Discussion.

Answer: We appreciate the editor's insightful suggestion. Based on this comment, we then omitted several new figure references in the discussion section. Please kindly refer to our discussion.

Conclusion:

  1. Figure 8 cannot be included in this Section. The authors should limit themselves to briefly indicate the final message.

Answer: We appreciate for receiving a positive comment from the reviewer. Regarding the schematic figure 8, we integrated this figure with Discussion part (line 467-468). Furthermore, we also summarized the conclusion into a more compact conclusion. Therefore, please kindly refer to our conclusion section in line 547 to 551.

Conclusion

This study provides novel insights of USP6NL/EGFR axis that suppresses anticancer therapeutic responses, induces metastasis, and facilitates reduced sensitivity to TMZ treat-ment in patients with GBM in an autolysosome-dependent manner. Therefore, controlling the USP6NL may offer an alternative but efficient therapeutic strategy for targeting and eradicat-ing erstwhile resistant and recurrent phenotypes of aggressive GBM cells.

Minor criticisms

  1. Line 288. The acronym USL6NL is misspelled.

Answer: We greatly appreciate the attentive suggestion of the reviewer. According to this misspelling, we then replaced the acronym into a correct one; that is USP6NL. Therefore, please kindly refer to line 315.

  1. Line 442. The authors state that. “We demonstrated that overexpression of USP6NL was strongly correlated with EGFR in the patient.” This sentence has no sense: It is not in the patient, but in the tumor.

Answer: Thank you for the helpful suggestions from the reviewers. Indeed, we agreed with this comment that EGFR expression was actually evidenced in the GBM tumor tissue, not in the patient. Therefore, we re-wrote the sentence into a correct one. Please kindly refer to line 489 to 491 in Discussion section.

As shown in schematic Figure 8, this study first provides the novel insights into the role of USP6NL/EGFR in GBM-TMZ resistance. We demonstrated that overexpression of USP6NL was strongly correlated with EGFR in the GBM tumor tissue and cell lines.

  1. Line 462. It is said that “We further demonstrated the relationship between histone H2AX phosphorylation and the sequestration of DNA repair factors [44].” It is confusing because it could be interpreted that the referenced article was produced by the authors of the present manuscript, but it is not (Gottlieb, R.A.; Andres, A.M.; Sin, J.; Taylor, D.P.J. Untangling autophagy measurements: All fluxed up. Circ Res 2015, 116, 624).

Answer: It was a pleasure to receive very constructive feedback from the reviewer. Indeed, it seems that we have misplaced this reference for next following sentence. Therefore, we deleted this citation and replaced into a correct one. Please kindly refer to line 516-526 in Discussion section.

We further demonstrated the relationship between histone H2AX phosphorylation and the sequestration of DNA repair factors. On the introduction of DSBs, the first cellular re-sponse is H2AX phosphorylation [46]. Colocalization of  BRCA1-Y-H2AX-RAD51 is critically important as DNA-binding factor during DSB [46]. Therefore, we evaluated BRCA1-Y-H2AX-RAD51 colocalization and observed a strong correlation between USP6NL and BRCA1-yH2AX-RAD51 in the TCGA-GBM data sets. Furthermore, expression of phosphorylated H2AX (y-H2AX) was reduced in shRNA-mediated knockdown of USP6NL cells (shUSP6NL), and TMZ-resistant cells demonstrated higher expression and colocalization of y-H2AX-BRCA1-RAD51, suggesting that TMZ induced DNA repair. Furthermore, the expression of DNA repair factors was decreased in shUSP6NL GBM cells, highlighting the key role of the USP6NL axis in the DNA damage repair response.

  1. Lines 453-55. The same about “Expression of stem cell transcription factors [42], CD44, SOX2, CD133 ABCG2, and NONG, together with USP6NL, was induced in TMZ-resistant GBM cells. This suggests a strong correlation between USP6NL level and self-renewal process (Figure 3).” This sentence is confusing. What is the reason to quote this article? (Bao, B.; Ahmad, A.; Azmi, A.S.; Ali, S.; Sarkar, F.H. Overview of cancer stem cells (cscs) and mechanisms of their regulation: Implications for cancer therapy. Curr Protoc Pharmacol 2013, Chapter 14, Unit-14.25.)

Answer: We appreciate the editor's insightful suggestion. According to this comment, we have then replaced the sentence into a sentence with clearer meaning and appropriate citation. Please kindly refer to line 505-515 in Discussion section.

We demonstrated that TMZ-resistant GBM cells harbor the ability of self-renewal, to gen-erate tumorspheres, to invade, and to enhance the EMT process. Following those phenotypes, it has been described that the expression of stem cell transcription factors such as CD44, SOX2, CD133 ABCG2, and Nanog, contribute a pivotal role for determining aggressiveness of cancer [43]. Therefore, we observed those stemness markers along with USP6NL were induced in TMZ-resistant GBM cells. This suggests a strong correlation between USP6NL level and self-renewal process. Furthermore, EMT-associated markers (N-cadherin upregulation of E-cadherin downregulation) were modulated in TMZ-resistant cells, confirming the EMT transi-tion of GBM [44,45]. Furthermore, the shRNA-mediated knockdown of USP6NL resulted in reduced expression of the aforementioned markers, thus confirming the correlation between USP6NL and TMZ resistance.

  1. Line 467. It is said that “Furthermore, y-H2AX expression was reduced in shUSP6NL cells”. In the case of abbreviations, please indicate what they are stand for.

Answer: We impressively appreciate the helpful suggestion of the reviewer. According to this comment, we then added the explanation of those abbreviations in new sentence. Please kindly refer to line 516-525 in Discussion section.

We further demonstrated the relationship between histone H2AX phosphorylation and the sequestration of DNA repair factors. On the introduction of DSBs, the first cellular re-sponse is H2AX phosphorylation [46]. Colocalization of  BRCA1-Y-H2AX-RAD51 is critically important as DNA-binding factor during DSB [46]. Therefore, we evaluated BRCA1-Y-H2AX-RAD51 colocalization and observed a strong correlation between USP6NL and BRCA1-yH2AX-RAD51 in the TCGA-GBM data sets. Furthermore, expression of phosphorylated H2AX (y-H2AX) was reduced in shRNA-mediated knockdown of USP6NL cells (shUSP6NL), and TMZ-resistant cells demonstrated higher expression and colocalization of y-H2AX-BRCA1-RAD51, suggesting that TMZ induced DNA repair. Furthermore, the expression of DNA repair factors was decreased in shUSP6NL GBM cells, highlighting the key role of the USP6NL axis in the DNA damage repair response.

Conclusion

Finally, I consider that it is an interesting and original work that gives relevant information about a possible effect of USP6NL on GBM cells and that it could contribute to a better knowledge of the role of this molecule in the oncogenesis and treatment of GBM. However, the authors should modify the text according to the reviewer considerations.

Thanks for reviewer's comments

Reviewer 2 Report

In the present article entitled: SP6NL and EGFR signaling axis regulates the ubiquitin-mediated DNA repair and temozolomide-resistance in glioblastoma, authors are presenting a study in which they claim that the results provide novel insights into the probable mechanism through which USP6NL/EGFR signaling might suppress anticancer therapeutic response, induce metastasis, and facilitate reduced sensitivity to temozolomide treatment in patients with GBM in an autolysosome-dependent manner. Therefore, controlling the USP6NL may offer an alternative but efficient therapeutic strategy for targeting and eradicating erstwhile resistant and recurrent phenotypes of aggressive GBM cells.

Comments:

The article could be taken into consideration only, but only after major revisions. The paper is full of overstatements not supported by scientific data, contradictions, and inaccuracies in the field of glioblastoma. The study should focus only on the role of USP6NL in TMZ resistance. The authors are claiming a theory on the possible interaction and role of these two genes as a duet without ever proving the real role of the two genes/proteins combined. The EGFR is somehow associated but its real role in combination with USP6NL is never shown or proven.

There are many comments that make the paper absolutely not acceptable in this form.

Comments:

1)                  In the abstract and in the introduction as well as in the conclusions, the authors talk about “metastasis” which really sounds screechy when it refers to glioblastoma which is known to not metastasize. There are really very rare cases, but t is not something that should be mentioned when you are an expert in the field.

2)                  In the abstract authors claim that the EGFR mutated form is associated with shorter survival (line 26) and say the opposite in line 61. These are inaccuracies, not needed. The EGFR is amplified many times in glioblastoma but it is considered nowadays so appealing as a therapeutic target.

3)                  The introduction has to be re-written, there is not a real rationale behind it, it is messy and different topics are introduced randomly going back and forth through them.  The association with EGFR is forced and it doesn’t have a real directionality. It seems like the author has stepped luckily into the USP6NL gene somehow and is trying to give a logical reason for studying it. The story could hold as strongly, if not better, only on USP6NL, without having to connect it to EGFR.

4)                  In the self-made resistant GB cells, is a concentration of 100uM TMZ used always to maintain the cells resistant? Does therefore TMZ at that concentration become a usual ingredient of the cell medium?

5)                  In line 138 can the authors specify the number of TCGA patients tested? On line 143 who are these samples? When were they used in the paper? how many are they exactly?

6)                  Are the U87 and T98G Resistant cells transfected with shRNA? Explain line 148.

7)                  Line 154, when do you show in the results 24 or 48hr cell results?

8)                  Line 177, which antibodies?

9)                  Line 210 where are these experiments?

10)              Figure 1A;B shows that these two genes are overexpressed in GB compared to controls but it doesn’t imply that there is a functional association between them, there are thousands of other genes that behave in the same way and that could be associated to USP6NL. Moreover, the correlation in Figure 1c is not significant, in the text at line 267 the authors report a p<0.05 but the p is 0.12.

11)              Figure 1d is not an immunofluorescent assay as reported on line 270. Moreover, what tissues are these, where do they come from and what do authors mean by nontumor counterpart?

12)              At line 278, a huge overstatement is reported, these results have nothing to do with a prognostic marker.

13)              The use of 4 cell lines is not necessary. Cell lines HB683 which are not glioblastoma cells and DBTRG05MG give useless information.

14)              At line 288 again experiments that are following are not proving the crucial role of EGFR in TMZ resistance but only of USP6NL. Actually, these experiments are for the moment only showing that when you create resistant TMZ cells USP6NL is simply increasing its expression. A more important role in resistance to TMZ is expressed with the experiment in figure 2c and d but only refer to USP6NL.

15)              Figuer2f  are not scatterplots, but histograms, these do not look to me as scatter plots of flow cytometry experiments.

16)              AInline 316, the title is wrong since in this paragraph authors are just showing that TMZ resistant cells have changed self-renew capabilities. Here the gene USP6NL is not artificially modified yet. Also at line 327 again the title is wrong since UPS6NL is not modified, and the authors are just showing again in figure 3D that the protein, as shown in Figure 1a, is over-expressed in resistant cells, as well as the other markers of stemness. Why they didn’t perform any test of stemness marker expression with shUSP6NL?

17)              In line 333, the title is completely wrong since in these experiments the authors never perform an experiment treating cells with TMZ. Here unlike earlier, they prove that by modifying USP6NL expression there is a modification of the stemness genes but nothing related to TMZ action. Again title in line 349 is wrong.

18)              Paragraph at line 356: Here the authors should explain better the results, which are interesting since modulating the USP6NL they have variations of the different genes, even though the correlations are not so significant as they claim. See Figure5 a. Moreover, some figures are repeated many times 5b, 5c and 5d are showing the same results, the three histogram graphs should go together.

19)              In figure 6 correlations look very weak to me, the title of every single figure should be more correct since it is not clear what they mean by control, are these resistant cells scramble?

20)              In figure 6b, the TMZ lane should come second, after the control, and it should be reported the concentration. Moreover please explain why EGFR protein level decreases when cells are treated with TMZ alone?

21)              Explain better what it means in figure 6F. experiment 6d,e,f represents a separate experiment from the rest of the figure, that should have its own paragraph.

22)              Figure 7 c cannot be appreciated at all. On what day these tumors were examined? This experiment seems well done and interesting.

23)              The discussion is full of inaccuracies, and results are overstated and should be re-written with another aim. The conclusions are not supported by the data.

Author Response

# New Comments from Reviewer 2

  1. In the abstract and in the introduction as well as in the conclusions, the authors talk about “metastasis” which really sounds screechy when it refers to glioblastoma which is known to not metastasize. There are really very rare cases, but it is not something that should be mentioned when you are an expert in the field.

Answer: We excitingly appreciate the very constructive suggestion of the reviewer. Indeed, after digesting this comment, we decided to change the “metastasis” term into “invasion” or “invasiveness”, as in our key result the USP6L/EGFR axis could modulate invasion, migration, EMT, and cancer stemness phenotype of glioblastoma that those phenotypes remark the invasiveness trait of cancer. Similarly, this comment also in line with recent review that glioblastoma would never undergo metastasis extra-cranially. However, invasion of GBM may frequently occur to surrounding structures, such as vasculatures, white matter tracts, and subarachnoid space that is orchestrated by specialized cancer cells subset [5]. Please kindly refer to updated abstract, line 404, and corrected conclusion.

Corrected Abstract: … The USP6NL level together with EGFR expression in human GBM tissue samples and cell lines associated with therapy resistance, tumor growth, and cancer invasion were investigated. … Herein, we find that deubiquitinase USP6NL and growth factor receptor EGFR were strongly associated with the oncogenicity and resistance of GBM both in vitro and in vivo toward temozolomide, as evidenced by enhanced migration, invasion, and acquisition of a highly invasive and drug-resistant phenotype by the GBM cells. … Our results provide novel insights into the probable mechanism through which USP6NL/EGFR signaling might suppress anticancer therapeutic response, induce cancer invasiveness, and facilitate reduced sensitivity to temozolomide treatment in patients with GBM in an autolysosome-dependent manner. …

Corrected Line 402-403: This finding suggests that targeting and inhibiting USP6NL can lead to the reduction of tumor initiation, invasion, drug resistance, and disease relapse in GBM.

Corrected Line 554-558: This study provides novel insights of USP6NL/EGFR axis that suppresses anticancer therapeutic responses, induces cancer invasiveness, and facilitates reduced sensitivity to TMZ treatment in GBM on an autolysosome-dependent manner.

  1. In the abstract authors claim that the EGFR mutated form is associated with shorter survival (line 26) and say the opposite in line 61. These are inaccuracies, not needed. The EGFR is amplified many times in glioblastoma but it is considered nowadays so appealing as a therapeutic target.

Answer: We thank the editor for the insightful suggestion. According to this suggestion, we would like to omit any contradiction by deleting words in line 26 that refer to “shorter survival in EGFR mutated patient” while preserving the following sentence: “role of EGFR in the survival of patients with GBM remains controversial” in line 62 of introduction section. We choose this option because in this study we did not evaluate the survival outcome or Kaplan-Meier curve due to our data limitation. However, in our speculation, the key data presented in figure 1 and 2 at least suggested that USP6NL/EGFR axis may determine drug resistance and invasiveness of GBM, which could signal the poorer outcome of GBM patients due to activation of this axis. Therefore, please kindly refer to updated abstract.

Corrected Abstract: Glioblastoma multiforme (GBM) is the most malignant glioma, with 30%–60% epidermal growth factor receptor (EGFR) mutation. This mutation is associated with unrestricted cell growth and increases the possibility of cancer invasion. Patients with EGFR-mutated GBM often develop resistance to the available treatment modalities and higher recurrence rates. The drug resistance observed is associated with multiple genetic or epigenetic factors.

  1. The introduction has to be re-written, there is not a real rationale behind it, it is messy and different topics are introduced randomly going back and forth through them. The association with EGFR is forced and it doesn’t have a real directionality. It seems like the author has stepped luckily into the USP6NL gene somehow and is trying to give a logical reason for studying it. The story could hold as strongly, if not better, only on USP6NL, without having to connect it to EGFR.

Answer: We impressively appreciate the helpful suggestion of the reviewer. According to this suggestion, we have already re-written some part of introduction section and also add more reference (ref no. 27) to highlight the important connection of USP6NL and EGFR trafficking. In this case, several previous evidences already noted USP6NL might delay EGFR endocytosis that result into upregulation of this oncogenic receptor in certain cancers, for example in colorectal and breast cancer. However, unavailability of this evidence in glioblastoma attracted our team to disclose more about this phenomenon whether also happened in GBM and how far the interaction of this axis could affect aggressive phenotype of GBM, particularly in determining drug resistance and cancer invasiveness. Therefore, please kindly refer to our reconstructed introduction section in line 105~128.

Corrected Introduction: Ubiquitin-specific peptidase 6 N-terminal like (USP6NL, also called RN-tre or Tre2) has been implicated in some human malignancies, namely gastric cancer, colorectal carcinoma, and breast cancer [23-25]. USP6NL acts as a GTPase-activating protein for Ras-related protein, and is involved in receptor trafficking. It inhibits EGFR internalization in a complex with EPS8 and decreases Rab5 activity [25]. A substantial amount of evidence supports the finding that USP6NL inhibits EGFR endocytosis. USP6NL increases the constitutive internalization of EGFR and α5β1  integrins, which suggests a role for USP6NL in the EPS8-constrained endocytosis of EGFR as well as the unstable organization of adhesion complexes and dissociation of the EGF-dependent adhesion complex [25,26]. Breast cancer cells with high levels of USP6NL experienced delayed endocytosis and degradation of EGFR, activated AKT signaling. Deficiency of USP6NL caused downregulation of EGFR, resulting in suppression of AKT and GLUT1 degradation and impairment of cellular proliferation [27]. Thus, we hypothesized that USP6NL regulates not only EGFR trafficking but also the cellular survival capacity or bio-availability of adhesion and growth factor receptors, including EGFR. Consistent with suggestion that targeting USP6NL enhanced β-catenin ubiquitination, suppressed cancer cell proliferation, and induced cell cycle arrest in colorectal cell lines [23], as well as considering that EGFR is the most commonly mutated driver of oncogenicity in patients with GBM, unfortunately, targeting EGFR with inhibitors has no or very less response toward the treatment [28].

New Reference:

  1. Avanzato, D.; Pupo, E.; Ducano, N.; Isella, C.; Bertalot, G.; Luise, C.; Pece, S.; Bruna, A.; Rueda, O.M.; Caldas, C.; et al. High USP6NL Levels in Breast Cancer Sustain Chronic AKT Phosphorylation and GLUT1 Stability Fueling Aerobic Glycolysis. Cancer Research 2018, 78, 3432-3444, doi:10.1158/0008-5472.CAN-17-3018.

  1. In the self-made resistant GB cells, is a concentration of 100uM TMZ used always to maintain the cells resistant? Does therefore TMZ at that concentration become a usual ingredient of the cell medium?

Answer: We thank the editor for this critical question. Our protocol for generating TMZ-resistant GBM cell line was previously described and modified from Ujifuku et al [1] and Zhang et al. [2], where the cells were initially exposed to a stepwise increment of TMZ up to 100-150 µM to select the representative clone with high resistant phenotype. Subsequently, after the selection period, the GBM cell lines need to maintain the resistant phenotype by keeping the TMZ drug in the complete medium containing 100 µM of TMZ. In that case, we modified our complete medium by adding TMZ every time we replaced the new medium. However, the protocol of TMZ dose in maintenance period is diverse that may range from 10 – 300 µM according to one review [6], but in our in-house protocol, we selected 100 µM which was a relatively “intermediate” dosage to adequately maintain the resistant phenotype. Therefore, please kindly refer to our methods section 2.1.

Corrected sentence: TMZ-resistant U87MGR and U251R cells were generated using a TMZ dose-escalation method up to 150 µM and then maintained at 100 µM TMZ for in vitro and in vivo experiments according to earlier reference [29,30].

New Reference:

  1. Ujifuku, K.; Mitsutake, N.; Takakura, S.; Matsuse, M.; Saenko, V.; Suzuki, K.; Hayashi, K.; Matsuo, T.; Kamada, K.; Nagata, I.J.C.l. MiR-195, miR-455-3p and miR-10a∗ are implicated in acquired temozolomide resistance in glioblastoma multiforme cells. 2010, 296, 241-248.
  2. Zhang, J.; Stevens, M.F.G.; Laughton, C.A.; Madhusudan, S.; Bradshaw, T.D. Acquired Resistance to Temozolomide in Glioma Cell Lines: Molecular Mechanisms and Potential Translational Applications. Oncology 2010, 78, 103-114, doi:10.1159/000306139.

  1. In line 138 can the authors specify the number of TCGA patients tested? On line 143 who are these samples? When were they used in the paper? how many are they exactly?

Answer: We thank the editor for this important question. To address this question, in result subsection, we previously mentioned that the total number of TCGA GBM patients that were included in this was 370 individuals. Therefore, to make it clearer, we added this sample size in our updated methods section. Accordingly, in our in-house Taipei Medical University-Shuang Ho Hospital GBM patient cohort, we already described the sample size including the basic clinical characteristics in our supplementary Table S1. This cohort would be employed for subsequent immunohistochemistry staining analysis which the result was provided in figure 1D. The cohort size was 60 individuals and already mentioned in figure legend 1D. Please kindly refer to our updated materals & methods subsection 2.2 Bioinformatic analysis of TCGA-GBM database.

2.2. Bioinformatic analysis of the TCGA-GBM database

TGCA-GBM mRNA expression data, along with clinical information were extracted and integrated with corresponding microarray data, including those of GBM and normal CNS tissues (n  =  370). Extracted clinical profiles include patients’ age, sex, ethnicity, follow-up duration (days), endpoint/event, method of initial confirmed diagnosis, histological type, EGFR mutation status, pathological stage, and grade. The study was approved by the Joint Institutional Review Board (JIRB) of the Taipei Medical University –Shuang Ho Hospital (Approval no.: JIRB N202101069). A total of 60 tissue samples from patients with primary and recurrent GBM were obtained from the Taipei Medical University-Shuang Ho Hospital GBM cohort and complied with the recommendations of the Declaration of Helsinki for Biomedical Research. Tissue samples were employed for further immunohistochemistry staining and analysis. Supplementary Table S1 showed the baseline clinical characteristics and outcome of GBM patients in SHH-GBM cohort.

  1. Are the U87 and T98G Resistant cells transfected with shRNA? Explain line 148.

Answer: We thank the editor for this vital question. Indeed, in our methods we would like to transfect both parental U87 and T98G and also the TMZ-resistant clone of U87 and T98G cell lines (we labeled them as U87MG-R and T98G-R). Therefore, to clarify this issue, we added new information regarding the resistant clone cells in our shRNA methods. Please kindly refer to our updated methods section, in subsection 2.3 shRNA transfection.

2.3. shRNA transfection of GBM cells

Both parental and resistant clones of U87MG and T98G cells were transfected with shRNA specifically targeting USP6NL or control/scramble shRNA purchased from Santa Cruz Biotechnology (Santa Cruz, CA, USA). The U87MG and T98G cells were transfected with shRNA following the manufacturer’s instructions. shRNA-USP6NL-transfected clones were then expanded for future use.

  1. Line 154, when do you show in the results 24 or 48hr cell results?

Answer: We thank the editor for this essential question. The result of CCK-8 viability assay was provided in figure 2B and 2D. Please kindly refer to our figure 2B and D.

Figure 2. USP6NL is highly expressed in temozolomide (TMZ)-resistant GBM (U87MG-R and T98G-R) cells. (B) Viability assay performed using the CCK-8 assay kit showing that estab-lished TMZ-resistant cell lines, U87MG-R and U251-R, exhibited increased resilience toward TMZ treatment. (D) Reduction in the cell viability of TMZ-resistant GBM cells, demonstrating the effect of USP6NL knockdown sensitized resistant cells toward TMZ.

  1. Line 177, which antibodies?

Answer: We thank the editor for this critical question. To address this question, we already updated new detailed information regarding primary antibodies in this materials section. Please kindly refer to our updated subsection 2.7 Immunufluorescence staining methods.

2.7. Immunofluorescence staining

Representative human glioblastoma cell lines harboring either scramble control or knockdown of USP6L (shUSP6L) were plated in six-well chamber slides for 24 h to perform immunofluorescence analysis. The cells were fixed with 2% paraformaldehyde and probed with primary antibodies. To determine the positive signal, a fluorophore-conjugated secondary antibody was added following examination using a Zeiss Axiophot (Carl Zeiss) fluorescence microscope. The nuclei of viable cells were detected through 4′,6-diamidino-2-phenylindole (DAPI) staining. The primary antibodies being used were purchased from several different companies and diluted under specific concentrations, such as E-cadherin (1:100, #3195, rabbit mAb; Cell Signaling), N-cadherin (1:100, #13116, rabbit mAb; Cell Signaling), yH2AX (1:100, #7631, rabbit mAb; Cell Signaling), RAD51 (1:100, ab63801, rabbit polyclonal-Ab; Abcam), BRCA2 (1:100, # PA5-96128, rabbit polyclonal-Ab; ThermoFisher), ATG5 (1:100, ab228668, rabbit polyclonal-Ab; Abcam), ATG7 (1:100, ab133528, rabbit mAb; Abcam), Beclin-1 (1:100, ab62557, rabbit polyclonal-Ab; Abcam), and LC3A/B (1:100, #4108, rabbit polyclonal-Ab; Cell Signaling). The negative controls were carried out by omitting the primary antibody.

  1. Line 210 where are these experiments?

4C

3C

Answer: We thank the editor for this important question. The result of cell invasion assay was provided in figure 3C and 4C. Please kindly refer to our figure 3C and 4C.

Figure (3C). Representative image of invasive properties of TMZ-resistant GBM cells (U87MG-R and T98G-R). (4C) Representative image of invasive properties of USP6NL-knockdown GBM cells (U87MG-R and T98G-R).

  1. Figure 1A;B shows that these two genes are overexpressed in GB compared to controls but it doesn’t imply that there is a functional association between them, there are thousands of other genes that behave in the same way and that could be associated to USP6NL. Moreover, the correlation in Figure 1c is not significant, in the text at line 267 the authors report a p<0.05 but the p is 0.12.

Answer: We thank the editor for this important comment. Indeed, in our approach, we would like to preliminarily investigate the co-association of both our interest gene (USP6NL and EGFR) in the publicly available transcriptomic dataset then confirmed in our in-house GBM patient cohort by performing immunohistochemistry staining. Even tough, lots of gene would have positive correlation to USP6NL, those myriad genes were not our scope to deliver which this study only focus on EGFR and USP6NL.Moreoever, we think that lots of study also used this approach by preliminarily observed the correlation then confirm into their clinical cohort specimens. For the p value, we have already re-calculated the pearson’s correlation coefficient along with p value which achieved significant positive correlation (r = 0.22, p value = 0.0035) as provided in our updated figure 1C. Appropriate result was also updated in the result sub section 3.1 to mention recently calculated correlation and its significance. Please kindly refer to our updated figure 1C.

Figure 1C. Correlation analysis of mRNA expression of EGFR and USP6NL in the TCGA-GBM database.

  1. Figure 1d is not an immunofluorescent assay as reported on line 270. Moreover, what tissues are these, where do they come from and what do authors mean by nontumor counterpart?

Answer: We appreciate the editor's insightful suggestion. We appeared to have misplaced this vital information. It is indeed result of immunohistochemistry staining. We have updated this information in the result section, subsection 3.1. The tissue specimens were obtained from our Taipei Medical University-Shang Ho Hospital (TMU-SHH) GBM patient cohort as mentioned in methods section, subsection 2.2. The non-tumor counterpart means the adjacent area of tumor part that histologically resembles normal tissue adjacent to the tumor (NAT) area as this part is commonly used as a control in cancer studies. Please kindly refer to our updated result, subsection 3.1.

Corrected sentence: Our immunohistochemistry analysis of our own in-house SHH-GBM cohort (n = 60) revealed enhanced USP6NL expression as represented by high Q-score in the EGFR-amplified samples compared with their nonamplified or nontumor counterparts (nontumor << EGFRamp- << EGFRamp+) (Figure 1D). Non-tumor counterparts represented as histologically normal part in adjacent area of tumor portion (NAT).

Corrected figure legend: Figure 1D. Immunohistochemistry analysis of USP6NL expression in EGFR-amplified samples compared with their nonamplified SHH-GBM cohort (n = 60)

  1. At line 278, a huge overstatement is reported, these results have nothing to do with a prognostic marker.

Answer: We appreciate the editor's insightful comment. To reduce overstatement of this sentence, we already updated the prognostic marker into other statement that USP6NL may predict high expression or amplification of EGFR that may subsequently have different phenotype of glioma tumors. Please kindly refer to line 333-335.

Corrected sentence: Taken together, these results suggest USP6NL as an interesting potential to later predict EGFR overexpression or amplification in gliomas that may subsequently have different phenotype of glioma tumors.

  1. The use of 4 cell lines is not necessary. Cell lines HB683 which are not glioblastoma cells and DBTRG05MG give useless information.

Answer: We appreciate the editor's critical comment. In this case, we have already validated the profile of our in vitro cell line model according to public portal of DepMap. This established portal has deposited well-known CCLE database and contained diverse profile of numerous commercially available cancer cell line. According to DepMap portal, Hs683 and DBTRGO5MG are included in glioma lineage subtype, specifically oligodendroglioma and glioblastoma, respectively. Therefore, we still consider both Hs683 and DBTRGO5MG are still relevant for comparing diversity of USP6NL expression across glioma cell lines. We provided the link for DepMap portal of both cell lines’ profile as follow:

  • Hs683: https://depmap.org/portal/cell_line/ACH-000067?tab=mutation
  • DBTRGO5MG: https://depmap.org/portal/cell_line/ACH-000863?tab=mutation

  1. At line 288 again experiments that are following are not proving the crucial role of EGFR in TMZ resistance but only of USP6NL. Actually, these experiments are for the moment only showing that when you create resistant TMZ cells USP6NL is simply increasing its expression. A more important role in resistance to TMZ is expressed with the experiment in figure 2c and d but only refer to USP6NL.

Answer: We appreciate the editor's critical comment. In our study, we previously considered not to evaluate role of EGFR signaling for provoking temozolomide resistance because numerous studies has already signified the evidence of EGFR signaling activation contributed to TMZ chemoresistance in glioma [7-10]. Although not much strong, we could also see across from the in silico public dataset, clinical specimens, and in vitro cell line study the co-association of EGFR and USP6NL. However, even though EGFR was critical for generating TMZ resistance, inhibition this single EGFR signaling only resulted in a failure that require other approach such as combining with other oncogene blocking agent [11]. Therefore, in this study, we were curious to explore other approach of targeting EGFR in the context of EGFR trafficking that was putatively regulated by USP6NL and observed that modulation of this axis may also contribute for TMZ chemoresistance and cancer invasiveness in glioma. At this moment, we could not elaborate more by providing deeper and more detailed data as expected by reviewer except our current “straightforward” result regarding crucial role of EGFR.

  1. Figure 2f are not scatterplots, but histograms, these do not look to me as scatter plots of flow cytometry experiments.

Answer: We appreciate insightful suggestion from the reviewer. We appeared to have misplaced this vital information. Therefore, we have corrected “scatterplot” to “bar graph” in the figure legend 2F in response to your comment. Please kindly refer to our updated figure legend 2F.

Corrected figure legend: Figure 2F. Bar graph showed representative proportion of apoptotic cells according to flow cytometry of Annexin V-FITC/PI staining in USP6NL knockdown and control TMZ-resistant cells treated with TMZ.

  1. In line 316, the title is wrong since in this paragraph authors are just showing that TMZ resistant cells have changed self-renew capabilities. Here the gene USP6NL is not artificially modified yet. Also at line 327 again the title is wrong since UPS6NL is not modified, and the authors are just showing again in figure 3D that the protein, as shown in Figure 1a, is over-expressed in resistant cells, as well as the other markers of stemness. Why they didn’t perform any test of stemness marker expression with shUSP6NL?

Answer: We appreciate for a very constructive suggestion from the reviewer. Indeed, after we digest for this comment, we then corrected the title of this paragraph in sub section 3.3 to describe most key results in this paragraph that mainly showed the stemness properties of TMZ resistant glioma cells, including increased expression of USP6NL in this resistant clone. Similarly, we also made a correction regarding the title of figure legend 3 to represent most result provided in figure 3. Actually, we have already performed experiment to determine significant reduction of stemness marker in shUSP6NL of GBM resistant cells with the results were provided in figure 4 D, E, and F. Please kindly refer to our updated title sub section 3.3 and figure legend 3.

Corrected title: 3.3. Association of drug resistance with high USP6NL and cancer stemness properties

Corrected figure legend: Figure 3. USP6NL is markedly expressed in TMZ resistant cells with high stemness phenotypes

  1. In line 333, the title is completely wrong since in these experiments the authors never perform an experiment treating cells with TMZ. Here unlike earlier, they prove that by modifying USP6NL expression there is a modification of the stemness genes but nothing related to TMZ action. Again title in line 349 is wrong.

Answer: We are grateful to receive a very positive comment from the reviewer. After digesting this comment, we then corrected the title of this paragraph in sub section 3.4 so that the title could describe key results in this section that mainly showed reduction of stemness properties of TMZ resistant glioma cells after knock-down of USP6NL. Accordingly, we also modified the title of figure legend 4 in order to appropriately describe mot result provided in figure 4. Please kindly refer to our updated title sub section 3.4 and figure legend 4.

Corrected title: 3.4. Abrogation of USP6NL ameliorated the stemness phenotype of GBM resistant cells

Corrected figure legend: Figure 4. USP6NL inhibition regulates cancer stemness properties of GBM cells

  1. Paragraph at line 356: Here the authors should explain better the results, which are interesting since modulating the USP6NL they have variations of the different genes, even though the correlations are not so significant as they claim. See Figure5 a. Moreover, some figures are repeated many times 5b, 5c and 5d are showing the same results, the three histogram graphs should go together.

Answer: We are grateful to receive a critical comment from the reviewer. To address this comment, we already modified our strongly correlated into only positively correlated in order to not over claim the result provided in this figure. For other subsequent comment, we have our particular decision not to merge the graph together because in our story we would like to first describe in 5B that γ-H2AX foci formation as one of the earliest events detected in cells following double-strand break was markedly reduced in USP6NL. Then, after describing much more common marker of DNA damage as represented by γ-H2AX foci formation, we then tried to introduce new molecular targets for sensitizing glioma cells to TMZ and are associated with DNA damage response; those were RAD51 and BRCA2. In figure 5C we would like to provide lower resolution to understand the fluorescence pattern modulation in most cells, then in figure 5D we highlighted more in detail the reduction of those markers under knock-down of USP6NL. Therefore, this sequence of story was already constructed as it is in the section 3.5 and considered to represent most of our concept of how USP6NL might modulate the DNA Damage Response in glioma cells. Re-organizing this graph by merging those figures might collapse our main idea and the story layout of this section. Thus, we apologize not to consider this comment at this moment.

  1. In figure 6 correlations look very weak to me, the title of every single figure should be more correct since it is not clear what they mean by control, are these resistant cells scramble?

Answer: We are grateful to receive a critical comment from the reviewer. To address this comment, we already re-organized label in our figure 6B to make it more understandable the control which was indeed basically the shScramble of resistant cells. Please kindly refer to our updated figure 6B.

       Figure 6B. Immunofluorescence image of the expression of autophagic markers after USP6NL inhibition and TMZ treatment

  1. In figure 6b, the TMZ lane should come second, after the control, and it should be reported the concentration. Moreover please explain why EGFR protein level decreases when cells are treated with TMZ alone?

Answer: We are thankful to obtain vital comment from the reviewer. According to this comment, we have since re-organized the dosage label in our figure 6B to make it more comprehensible that TMZ dosage was 200 µM as we also mentioned in the section 3.6. For the next question, in figure 6C, indeed our result was consistent with previous study that demonstrated TMZ treatment would down-regulate EGFR expression in chemo-resistant clone, and this de-activation of EGFR under TMZ treatment might indicate specific subset with high chemo-resistance that was less sensitive to anti-EGFR drug [12]. Please kindly refer to our updated figure 6B.

Figure 6B. Immunofluorescence image of the expression of autophagic markers after USP6NL inhibition and TMZ treatment

  1. Explain better what it means in figure 6F. experiment 6d,e,f represents a separate experiment from the rest of the figure, that should have its own paragraph.

Answer: We are appreciative to obtain such constructive comment from the reviewer. At this moment we apologize not to consider for separating this figure into two distinct plots due to restriction of information that may further limit the content of paragraph if this paragraph was separated into two. Accordingly, for next suggestion, we have already re-written the result in section 3.6 to better explain the result in figure 6F. Please kindly refer to our updated section 3.6.

Corrected sentence: Thus, we examined that loss of USPNL markedly enhanced ubiquitination activity of EGFR in GBM cells transfected with shUSP6NL compared with the control (NT) (Figure 6F), and the results indicated that USP6NL may play a vital role in the protection and accumulation of EGFR by promoting its deubiquitination and preventing from subsequent downregulation.

  1. Figure 7c cannot be appreciated at all. On what day these tumors were examined? This experiment seems well done and interesting.

Answer: We are grateful to acquire this question from reviewer. This experiment was done after around 3 weeks (day 22) of subcutaneous inoculation to determine apoptosis induction by TMZ and loss of USP6NL. To make it clearer, we have updated the figure legend and added tumor observation at day 22 as the time point of this experiment. Please kindly refer to our updated figure legend of figure 7.

Corrected figure legend: Figure 7C. Apoptosis analysis in the tumor cells (day 22) by using the TUNEL assay

  1. The discussion is full of inaccuracies, and results are overstated and should be re-written with another aim. The conclusions are not supported by the data.

Answer: We are grateful to acquire this comment from reviewer. However, we could not understand this comment and hard to address this suggestion, specifically we did not know in what part of discussion that was inaccurate and based on what obvious reason. In conclusion part, we think that each statement was already supported with the result of this study. In that case, we would like to clarify, in our conclusion we stated: “This study provides novel insights of USP6NL/EGFR axis that (1) suppresses anticancer therapeutic responses, (2) induces cancer invasiveness, and (3) facilitates reduced sensitivity to TMZ treatment in GBM on an (4) autolysosome-dependent manner”. In the first point, this conclusion was based on several data such as figure 2 D (in vitro) and figure 7 A (in vivo) that inhibition of USP6NL could enhance response to TMZ. The association of USP6NL and EGFR as one axis including their interaction was then clarified clinically in figure 1D and 6 D, E, F (consisting of USP6NL-EGFR immunoprecipitation and ubiquitination of EGFR). Second point of our conclusion was supported by several data such as figure 4 A-F in which this figure mainly highlights attenuation of cellular invasion and cancer stemness markers following loss of USP6NL in the resistant clone of GBM cells. The third point was quite similar to first point, in which this conclusion was based on several data such as figure 2 D (in vitro) and figure 7 A (in vivo) that USP6NL deficiency could lead to enhance sensitivity to TMZ. Our last point in conclusion was supported mainly by figure 6 B and C that inhibition of USP6NL could induce autophagy that its process was marked by autolysosome formation. Thus, according to that reason, we apologize not to consider this comment at this moment.

References

  1. Ujifuku, K.; Mitsutake, N.; Takakura, S.; Matsuse, M.; Saenko, V.; Suzuki, K.; Hayashi, K.; Matsuo, T.; Kamada, K.; Nagata, I.J.C.l. MiR-195, miR-455-3p and miR-10a∗ are implicated in acquired temozolomide resistance in glioblastoma multiforme cells. 2010, 296, 241-248.
  2. Zhang, J.; Stevens, M.F.G.; Laughton, C.A.; Madhusudan, S.; Bradshaw, T.D. Acquired Resistance to Temozolomide in Glioma Cell Lines: Molecular Mechanisms and Potential Translational Applications. Oncology 2010, 78, 103-114, doi:10.1159/000306139.
  3. Lopez-Gines, C.; Gil-Benso, R.; Ferrer-Luna, R.; Benito, R.; Serna, E.; Gonzalez-Darder, J.; Quilis, V.; Monleon, D.; Celda, B.; Cerdá-Nicolas, M. New pattern of EGFR amplification in glioblastoma and the relationship of gene copy number with gene expression profile. Modern Pathology 2010, 23, 856-865, doi:10.1038/modpathol.2010.62.
  4. Detre, S.; Saclani Jotti, G.; Dowsett, M. A "quickscore" method for immunohistochemical semiquantitation: validation for oestrogen receptor in breast carcinomas. Journal of Clinical Pathology 1995, 48, 876-878, doi:10.1136/jcp.48.9.876.
  5. Vollmann-Zwerenz, A.; Leidgens, V.; Feliciello, G.; Klein, C.A.; Hau, P. Tumor Cell Invasion in Glioblastoma. International journal of molecular sciences 2020, 21, 1932, doi:10.3390/ijms21061932.
  6. Lee, S.Y. Temozolomide resistance in glioblastoma multiforme. Genes & Diseases 2016, 3, 198-210, doi:https://doi.org/10.1016/j.gendis.2016.04.007.
  7. Guo, G.; Gong, K.; Puliyappadamba, V.T.; Panchani, N.; Pan, E.; Mukherjee, B.; Damanwalla, Z.; Bharia, S.; Hatanpaa, K.J.; Gerber, D.E.; et al. Efficacy of EGFR plus TNF inhibition in a preclinical model of temozolomide-resistant glioblastoma. Neuro-Oncology 2019, 21, 1529-1539, doi:10.1093/neuonc/noz127.
  8. Ciechomska, I.A.; Gielniewski, B.; Wojtas, B.; Kaminska, B.; Mieczkowski, J. EGFR/FOXO3a/BIM signaling pathway determines chemosensitivity of BMP4-differentiated glioma stem cells to temozolomide. Experimental & Molecular Medicine 2020, 52, 1326-1340, doi:10.1038/s12276-020-0479-9.
  9. Tsai, Y.T.; Wu, A.C.; Yang, W.B.; Kao, T.J.; Chuang, J.Y.; Chang, W.C.; Hsu, T.I. ANGPTL4 Induces TMZ Resistance of Glioblastoma by Promoting Cancer Stemness Enrichment via the EGFR/AKT/4E-BP1 Cascade. Int J Mol Sci 2019, 20, doi:10.3390/ijms20225625.
  10. Yeom, S.-Y.; Nam, D.-H.; Park, C. RRAD Promotes EGFR-Mediated STAT3 Activation and Induces Temozolomide Resistance of Malignant Glioblastoma. Molecular cancer therapeutics 2014, 13, 3049-3061, doi:10.1158/1535-7163.MCT-14-0244.
  11. Meng, X.; Zhao, Y.; Han, B.; Zha, C.; Zhang, Y.; Li, Z.; Wu, P.; Qi, T.; Jiang, C.; Liu, Y.; et al. Dual functionalized brain-targeting nanoinhibitors restrain temozolomide-resistant glioma via attenuating EGFR and MET signaling pathways. Nature Communications 2020, 11, 594, doi:10.1038/s41467-019-14036-x.
  12. Areeb, Z.; Stuart, S.F.; West, A.J.; Gomez, J.; Nguyen, H.P.T.; Paradiso, L.; Zulkifli, A.; Jones, J.; Kaye, A.H.; Morokoff, A.P.; et al. Reduced EGFR and increased miR-221 is associated with increased resistance to temozolomide and radiotherapy in glioblastoma. Scientific Reports 2020, 10, 17768, doi:10.1038/s41598-020-74746-x.

Reviewer 3 Report

In their paper titled "USP6NL and EGFR signaling axis regulates the ubiquitin-mediated DNA repair and temozolomide resistance in glioblastoma", the authors investigated the role of the ubiquitin-specific protease 6 N-terminal-like protein (USP6NL) in glioblastoma resistance to temozolomide treatment. The present work is excellently elaborated, the experiments and the study are carefully designed to demonstrate the importance of USP6NL from the in vitro to the in vivo level. The results obtained show significant differences between cells resistant to temozolomide treatment and those in which USP6NL was knocked down. Discussion highlights the role of apoptosis and autophagy in the cell response and the proteins or genes involved in signal transduction. The manuscript is appropriate for the journal Biomedicines and will be of great interest to readers.

I have a few minor comments:

Page 5 line 203 the number of cells was not 1x105 cells.

The information in Figure 1 is not readable.

Page 7 line 304: Figure 2F does not show the sub-G1 population. Maybe the results are missing.

Even though the pictures are nicely prepared, the text in them is mostly written with too small font and not very readable.

Author Response

# Comments from Reviewer 3

  1. Page 5 line 203 the number of cells was not 1x105 cells.

Answer: We thank the editor for the insightful suggestion. According to this comment, we then revised the annotation of cell line number in new sentence. Please kindly refer to line 232 to 241 in subsection of Cell invasion assay.

Corrected Sentence: Chamber membranes (8 μm, BD Falcon) were not precoated with 6 µL Matrigel at 4 °C overnight before seeding with 2 × 104 cells.

  1. The information in Figure 1 is not readable.

Answer: Thank you for the helpful suggestions from the reviewers. According to this comment, we then improved the resolution and proportion size of figure 1 in order to enhance readability of this content. Please kindly refer our figure 1.

  1. Page 7 line 304: Figure 2F does not show the sub-G1 population. Maybe the results are missing.

Answer: We impressively appreciate the constructive suggestion of the reviewer. Indeed, figure 2F mainly described about cell death induction of temozolomide under knockdown of USP6L in resistant clone of GBM cell lines. Therefore, using annexin-V/PI assay we then evaluated percentage of viable, early, and late apoptosis to assess cytotoxicity following TMZ treatment. We apparently mentioned inaccurately of sub-G1 population in this figure. We also described our apoptosis assay that previously unmentioned in the method section. Therefore, we then replaced it into a more appropriate and representative sentence to describe the result in figure 2F. Please kindly refer to line 354 to 356.

Corrected Sentence: Enhanced TMZ-induced early and late of apoptosis, cytotoxicity, and reduction of viable cells were observed in both the USP6NL knockout U87MG-R and T98G-R cells (Figure 2F).

Round 2

Reviewer 1 Report

I consider that it is an interesting and original work that gives relevant information about a possible effect of USP6NL on GBM cells and that it could contribute to a better knowledge of the role of this molecule in the oncogenesis and treatment of GBM. I appreciate the modifications that the authors brought to their manuscript according to the reviewers´ comments. The authors have made several changes in the manuscript, which has resulted in an improved new version of the article.

Reviewer 2 Report

Accepted in present form